# A UNIVERSAL COMPRESSION THEORY FOR LOTTERY TICKET HYPOTHESIS AND NEURAL SCALING LAWS

**Hong-Yi Wang**
Princeton University & NTT Research
`hywang@princeton.edu`

**Di Luo**
Tsinghua University & UCLA
`diluo@tsinghua.edu.cn`

**Tomaso Poggio**
MIT
`tp@ai.mit.edu`

**Isaac L. Chuang**
MIT
`ichuang@mit.edu`

**Liu Ziyin**
MIT & NTT Research
`ziyinl@mit.edu`

## ABSTRACT

When training large-scale models, the performance typically scales with the number of parameters and the dataset size according to a slow power law. A fundamental theoretical and practical question is whether comparable performance can be achieved with significantly smaller models and substantially less data. In this work, we provide a positive and constructive answer. We prove that a generic permutation-invariant function of $d$ objects can be asymptotically compressed into a function of $\operatorname{polylog} d$ objects with vanishing error, which is proved to be the optimal compression rate. This theorem yields two key implications: (Ia) a large neural network can be compressed to polylogarithmic width while preserving its learning dynamics; (Ib) a large dataset can be compressed to polylogarithmic size while leaving the loss landscape of the corresponding model unchanged. Implication (Ia) directly establishes a proof of the *dynamical* lottery ticket hypothesis, which states that any ordinary network can be strongly compressed such that the learning dynamics and result remain unchanged. (Ib) shows that a neural scaling law of the form $L \sim d^{-\alpha}$ can be boosted to an arbitrarily fast power law decay, and ultimately to $\exp(-\alpha' \sqrt[m]{d})$.

## 1 INTRODUCTION

Training contemporary neural networks has become extremely costly. Modern models are very large and are often trained on enormous datasets. For example, GPT-4 is believed to have on the order of a trillion ($10^{12}$) parameters and to have been trained on roughly a trillion tokens. Training runs can occupy clusters comparable in scale to an entire data center. By contrast, the brain, a comparable biological computer, appears to require far less data. A rough back-of-the-envelope estimate illustrates the gap: suppose the auditory system of a person receives one word per second from birth. Then by age ten, a child would have heard about $10^8$ words, by which time most children have mastered their native language. This difference of roughly four orders of magnitude in data efficiency between artificial and biological systems suggests that current AI systems may not be using data optimally.

The data efficiency of large AI models is often summarized by neural scaling laws (NSL), in which the error $L$ decays approximately as a power of the dataset size (holding other factors fixed):

$$L(N) \propto N^{-\alpha}. \tag{1}$$

Empirical values of $\alpha$ for large language models typically lie between $0.1$ and $0.3$ (Kaplan et al., 2020). If we assume $\alpha = 0.1$ and that current models already achieve human-like language capability, then attaining the same capability with only $10^8$ tokens would require a modestly larger exponent, roughly $\alpha \approx 0.15$. Hence, even a small increase in the scaling exponent could substantially reduce training cost by bringing models closer to human-level data efficiency. Yet we currently lack principled guidance on whether, and by how much, the neural scaling laws can be improved.

In this work, we prove a universal result that, under suitable definitions of model width, enables subexponential compression (i.e., from $d$ objects to $\operatorname{polylog}(d)$) of essentially arbitrary neural networks

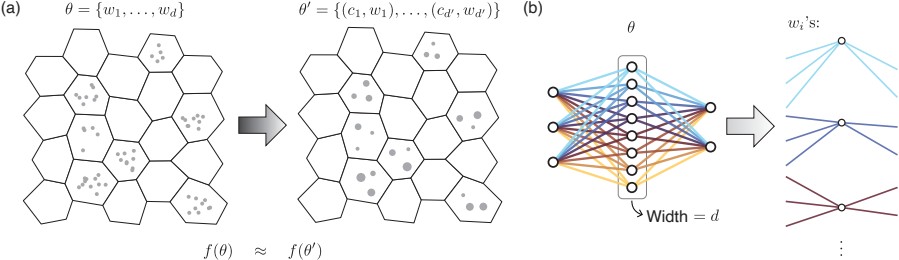

Figure 1: (a) Illustration of the main idea behind the compressibility of neural networks and datasets. (1) Permutation symmetry allows a high-dimensional function to be decomposed into a composition of $d$ low-dimensional "objects" (dots in the figure). (2) When $d$ is large, these objects become crowded, and those lying in denser regions are essentially redundant; they can be compressed into $d' = O(\text{polylog}\,d)$ objects. The potential curse of dimensionality can thus be mitigated, or even removed, when the underlying function is smooth—a lesson well known in nonparametric statistics. (b) Decomposing the linear weights of a neural network into "objects" of symmetric status.

and/or their training data, thereby opening the possibility of substantially improving a wide range of neural scaling laws. Our contributions are:

1. Proof of a universal compression theorem, showing by construction that almost any smooth symmetric function of $d$ elements can be compressed to a function with $O(\text{polylog}(d))$ elements losslessly (Section 4). Moreover, such a compression rate is optimal up to a constant factor.
2. Application to network compression, which leads to a proof of what we call the dynamical lottery ticket hypothesis (LTH), which states that a large network can be compressed such that its training dynamics is the same as the original (Section 5);
3. Application to compress datasets, which is a proof-of-concept showing that one can improve neural scaling laws significantly (Section 6).

Figure 1 provides a schematic of the main idea. All proofs appear in the Appendix; numerical experiments are presented alongside the related theoretical results.

## 2 PROBLEM SETTING

We begin with two motivating examples to illustrate the ubiquity of permutation symmetry in machine learning, and then introduce the permutation-symmetric functions we study.

**Data Permutation Symmetry.** The simplest form of permutation symmetry exists among data points. The loss function we minimize is $L(\theta, \{z_i\}_i^d)$, where $d$ is the number of data points, and $z_i = (x_i, y_i)$ is a data point consisting of input–label pairs. The loss function is the average of a per-sample loss $\ell$ over training data:

$$L = \frac{1}{d}\sum_{i=1}^{d} \ell(x_i, y_i, \theta) \equiv \mathbb{E}_z[\ell(x, y, \theta)]. \tag{2}$$

Because the loss function $L$ is a sum over the same function $\ell$ of each data point, permuting any pair of data points results in the same value of loss function. Moreover, since the dataset affects any trained model prediction via the gradient of the loss function, all model predictions are thus naturally permutation invariant.

**Neuron Permutation Symmetry.** Permutation symmetry in model parameter space depends on the structure of the model submodules. Consider the model $f(x) = W_2\sigma(W_1 x)$, where $W_1$ and $W_2$ are weight matrices, and $\sigma$ is an element-wise non-linearity. Let $w_i^T$ be the row vectors of $W_1$, and $v_i$ be the column vectors of $W_2$. Then the output reads

$$f(x) = \sum_{i=1}^{d} v_i\sigma(w_i^T x) \tag{3}$$

where $d$ is the output dimension of $W_1$, also the input dimension of $W_2$. $d$ is commonly known as the *width* of the neural network. The output is symmetric under the exchange of any pair of $(v_i, w_i) \leftrightarrow (v_j, w_j)$, another example of permutation symmetry. For a deep net, different layers can have multiple decoupled permutation symmetries. More generally, other modules that have

such permutation symmetry include fully connected layers, attention logits in self-attention, and attention outputs between different attention heads; it also exists when ensembling many models with the same architecture (Brea et al., 2019; Ziyin et al., 2025). Particularly, we further elaborate on the permutation symmetry in attention modules in Appendix F.

**General Permutation Symmetries.** Since our theory concerns compressing the objects while preserving the values of such symmetric functions, this framework is a unified approach to compressing either training data or any permutation-symmetric parameter set of a learning model. Generally, one can view the model or loss as a function of a set of permutation-symmetric inputs, while keeping other inputs implicit. We refer to each such input as an *object*, which may correspond to a data point or to a neuron weight ($w_i$). In this paper, each object is embedded in $m$ dimensions, and $d$ is the number of such objects. We primarily consider the limit $d \to \infty$.

**Definition 1.** *Let each $w_i \in V = \mathbb{R}^m$. $f : V^d \to \mathbb{R}$ is called a (permutation-)symmetric function in $\{w_i\}$ if, for any distinct $i, j \in [d]$, $f(\ldots, w_i, \ldots, w_j, \ldots) = f(\ldots, w_j, \ldots, w_i, \ldots)$.*

We also use $\theta = (w_1, \ldots, w_d)$ as a collective notation for all objects. To analyze the error induced by compression, we impose a mild regularity assumption on $f$. Specifically, it is known that any symmetric function admits a "deep set"–style universal representation (Zaheer et al., 2018) of the form

$$f(w_1, \ldots, w_d) = h\left(\sum_{i=1}^{d} g(w_i)\right), \tag{4}$$

where $h$ and $g$ are suitable functions. Importantly, for smooth $f$, one can choose $h$ and $g$ to be smooth as well (Tabaghi & Wang, 2023). We will use the following regularity assumption for the symmetric functions considered in this paper.

**Assumption 1.** *For all symmetric functions $f(\theta)$ studied in this paper, we assume that there exists a deep–set representation of $f$ as in Eq. (4), such that*

1. *Neither $h$ nor $g$ depends on $d$;*

2. *$h$ and $g$ are both Taylor-expandable with finite radii of convergence.*

While many ML models are non-smooth (e.g., ReLU networks), our compression results often extend to such settings at the level of conclusion (concretely, ReLU networks are studied in the numerical results in Figs. 3, 4 and 5), suggesting broader applicability beyond the analytic regime treated formally here.

**Notation.** Let $V = \mathbb{R}^m$ denote the space in which each object $w_i$ is embedded. $\otimes$ represents tensor product. The shorthand $[d]$ refers to the index set $\{1, 2, \ldots, d\}$, but when $x$ is a non-integer real number, $[x]$ denotes its closest integer. $S_d$ denotes the symmetric group of $d$ elements. For a nonnegative weight vector $\{c_i\}_{i=1}^d$, the support is defined as $\text{supp}(c_i) \equiv \{i \in [d] \mid c_i \neq 0\}$. For a set $S \subseteq V$, the diameter is $\text{diam}(S) \equiv \max_{x, x' \in S} \|x - x'\|$, where we use the Euclidean norm throughout this paper. $\mathcal{N}(\mu, \sigma^2)$ denotes the normal distribution with mean $\mu$ and variance $\sigma^2$. All other notations will be introduced in context.

## 3 RELATED WORKS

**Compression in AI.** Model and dataset compression has long been a central problem in AI (Han et al., 2015; Frankle & Carbin, 2018; Sorscher et al., 2022; Salomon, 2002; Wang et al., 2018). Yet, almost no theoretical framework exists to explain why, or to what extent, such compression is possible. A primary conceptual framework is the lottery ticket hypothesis (LTH) (Frankle & Carbin, 2018), which posits that within every network there exists a small subnetwork that, when retrained, can achieve the same performance as the original. Several theoretical works have established variants of the LTH (Malach et al., 2020; Pensia et al., 2021; da Cunha et al., 2022). However, these results typically fail to imply that the compressed model exhibits the same learning dynamics as the original—that is, that it reaches the same performance after training—which is arguably the most practical implication of the LTH. To date, the original formulation of the LTH remains unproven, precisely because of its dual requirement of both training and compression. We provide a more detailed discussion of this point in Sec. 5. Another closely related work is Ziyin (2024), which suggests the connection between symmetries and emergent sparsity during training.

**Neural Scaling Laws.** A major empirical guideline for training large language models (LLMs) is the neural scaling laws, which state that as the size of models and datasets increases, the generalization error decays as a power law: $L \propto d^{-\alpha}$, with $\alpha$ often small (Kaplan et al., 2020). Such small exponents pose a central obstacle for scaling LLMs. For example, when $\alpha = 0.1$, reducing the generalization error by half would require increasing the dataset size by a factor of 1000—an impractical demand given today's limited data availability. Sorscher et al. (2022) suggests the possibility of improving scaling laws through data pruning; however, their theory applies only to linear regression and assumes knowledge of the ground-truth model. Whether scaling laws can be improved in more general settings, and without requiring access to the ground truth, remains unknown.

## 4 UNIVERSAL COMPRESSION THEOREM

Compression is enabled by the observation that symmetric functions can be characterized by far fewer degrees of freedom than their apparent dimensions, which follows from a variant of the fundamental theorem of symmetric polynomials (FTSP). We then leverage this result to show that a family of compression algorithms ensures asymptotically lossless compression.

### 4.1 SYMMETRIC FUNCTIONS ARE COMPRESSIBLE

The value of a symmetric function does not depend on the specific ordering of $\theta = (w_1, \ldots, w_d)$, where each $w_i \in \mathbb{R}^m$. As a simple example, the joint probability distribution of $d$ i.i.d. random variables is symmetric in the sampled data points. This i.i.d. property underlies classical Shannon compression (Cover & Thomas, 2006). In this sense, permutation symmetry can be viewed as a natural generalization of independence (see, e.g., Bloem-Reddy & Teh (2020)), and it is thus natural that the compression of i.i.d. variables extends to the more general setting of permutation-symmetric variables. The following theorem shows that it suffices to keep track of the tensorial *statistical moments* of $\theta$, an idea traceable to Newton and Lagrange (Blum-Smith & Coskey, 2017). The following version of FTSP directly relates multivariate symmetric polynomials to their moments.

**Theorem 1** (Multivariate FTSP). *Let $\theta = (w_1, \ldots, w_d)$ with $w_i \in \mathbb{R}^m$. Any symmetric polynomial $f(\theta)$ (i.e., $f(\theta)$ is permutation invariant and is a polynomial of all scalar components $w_{i,a}$) can be expressed as a function of the moments $p_k$, $k \in [d]$, defined by*

$$p_k = \frac{1}{d} \sum_i w_i^{\otimes k} \equiv \frac{1}{d} \sum_i \underbrace{w_i \otimes \cdots \otimes w_i}_{k \text{ repetitions}}. \tag{5}$$

If $f(\theta)$ is a symmetric polynomial of degree $k$, then, $f(\theta)$ is fully determined by the first $k$ moments. Thus, any change to $\theta$ that preserves these moments leaves $f(\theta)$ unchanged. In other words, the variables can be compressed into the size of their leading $k$ moments as a linear space. The following theorem by Tchakaloff gives a direct upper bound on the number of elements required for the compression.

**Theorem 2** (Tchakaloff (1957)). *Let $\mu$ be a measure supported on $D \subset \mathbb{R}^m$. Then there exist $N$ points $w_j \in D$, with $N \leq N_{m,k} = \binom{m+k}{k}$, and positive weights $c_j$, such that the first $k$ moments are matched: $\forall l \in \{0, 1, \ldots, k\}$, $\int_D w^{\otimes l} d\mu(w) = \sum_{j=1}^N c_j w_j^{\otimes l}$.*

A proof follows from Carathéodory's theorem in dimension $N_{m,k}$ (Leonard & Lewis, 2015). Note that $N_{m,k}$ is precisely the dimension of the linear space of all moments up to order $k$ (including a fictitious dimension for $p_0$). Algorithm 2 in Appendix D guarantees such a compression: whenever there are more than $N_{m,k}$ weighted objects, we can always reduce the support to at most $N_{m,k}$ objects while preserving the first $k$ moments.

In the spirit of Tchakaloff's theorem, one can compress the original parameter set $\theta$ into a smaller set of weighted parameter set $\theta'$:

**Definition 2.** *A weighted parameter set $\theta'$ is defined as a collection of weight-parameter pairs*

$$\theta' = \{(c_1, w_1), (c_2, w_2), \ldots, (c_{d'}, w_{d'})\}, \tag{6}$$

*where each $c_j \geq 0$ and $w_j \in V = \mathbb{R}^m$. The moments of $\theta'$ and the values of symmetric functions are defined as an average over the discrete measure $\mu = \sum_{j=1}^{d'} c_j \delta_{w_j}$:*

$$p'_k = \frac{1}{\sum_{j=1}^{d'} c_j} \sum_{j=1}^{d'} c_j w_j^{\otimes k}, \quad f(\theta') = h\left(\sum_{j=1}^{d'} c_j g(w_j)\right), \tag{7}$$

where $f(\theta) = h(\sum_{i=1}^d g(w_i))$ *is the deep-set representation of a symmetric function.*

We remark that the original $\theta$ can also be viewed as weighted, with unit weight for each object. A feature of our compression is that it does not change any of the $w_i$'s but instead adjusts the weights so that they are supported on a smaller subset. When $f$ and $\{w_i\}_{i\in[d]}$ are fixed, we may regard the output as a function of only the weights $\{c_i\}_{i\in[d]}$. Specifically, given $f(\theta) = h(\sum_{i=1}^d g(w_i))$, we denote $\phi(c) = h\left(\sum_{i=1}^d c_i g(w_i)\right)$.

We are now ready to study approximating symmetric functions $f$ that are Taylor-expandable. Expanding $f$ to order $k$ yields a symmetric polynomial of degree $k$, and these moments can be matched with very few weighted objects. Consequently, the approximation error begins at order $(k+1)$.

**Theorem 3** (Moment matching in a small ball). *Let $f$ be a symmetric function acting on a weighted parameter set $\theta = \{(c_i, w_i)\}_{i\in[d]}$, and let $\phi$ be its corresponding function acting on weights. Let $r = \mathrm{diam}(\{w_i\}_{i\in[d]})$. Then, there exist nonnegative weights $\{c_i'\}_{i\in[d]}$ such that no more than $N_{m,k} \equiv \binom{m+k}{k}$ weights are nonzero, and*

$$|\phi(c') - \phi(c)| = O(dr^{k+1}). \tag{8}$$

Theorem 3 is a main ingredient of our compression theory. A lesson from this theorem is that dealing with a group of objects of small diameter is very effective in suppressing error, as we can choose a large enough $k$ to greatly suppress the compression error.

## 4.2 ASYMPTOTIC LOSSLESS COMPRESSION

Theorem 3 shows that compressing points in a small diameter enables a tight control on the error. By a sphere packing argument (see Lemma 1), if we put a lot ($d$) of objects in a finite region, then we can always find a subset of diameter $O((N/d)^{1/m})$ containing $N$ objects. This gives rise to a general strategy of compression in Algorithm 1, and any concrete algorithm that fits in this strategy is guaranteed to have vanishing compression error.

---

**Algorithm 1:** A compression algorithm family.

---

**Input** : a set of objects $\theta = \{w_i\}_{i\in d}$, target size $d'$, moment-matching order $k \in \mathbb{Z}_+$
**Output:** $\theta' = \{c_j, w_j\}_{j\in\mathrm{supp}(c)}$, where $|\mathrm{supp}(c)| \le d'$
Initialize $c_i = 1$ for all $i \in [d]$;
**while** $|\mathrm{supp}(c)| > d'$ **do**
    **Step 1 (clustering)**: Find a cluster $S \subseteq \mathrm{supp}(c)$ such that $|S| > N_{m,k}$ and
    $\mathrm{diam}\{w_j\}_{j\in S} = O(|\mathrm{supp}(c)|)^{-1/m}$ ;
    **Step 2 (moment matching)**: Update weights $\{c_j\}_{j\in S}$ such that they are supported on $N_{m,k}$
    objects, and the first $k$ moments are kept unchanged.
**end**

---

In terms of a practical compression algorithm, we first remark that Step 2 (moment matching) can be realized by Algorithm 2 in Appendix D, although several other moment-matching algorithms exist, especially in the case of peeling many points in one cluster (Piazzon et al., 2017). The most time-consuming part in Algorithm 2 is finding null vectors, so we estimate its time complexity as $dN_{m,k}^2$, which tells us that matching higher moments takes a much longer time. For clustering, ideally in each iteration one wants to greedily find the $(N_{m,k} + 1)$-subset with the smallest diameter, but the $k$-nearest neighbor problem is known to be NP hard in general. For clarity, in Theorem 4 of this section, we prove error bounds associated with the greedy strategy, indicating that a compression map satisfying our asserted error bounds universally exists. However, we perform numerical experiments using $k$-means clustering (see Appendix D), which works sufficiently well and is much faster.

The following theorem shows that with sufficiently large $k$ (compared to $m$, but still much smaller than $d$), one can compress the number of active objects to any power of $d$ with vanishing error.

**Theorem 4** (Universal Compression). *Let $\|w_i\| \le R$ for all $i \in [d]$. Algorithm 1 with moment-matching order $k$ can compress $\theta = \{w_i\}_{i=1}^d$ into $d' < d$ weighted objects $\theta'$, such that for any symmetric function $f$ satisfying Assumption 1,*

1. *The error is*

$$\mathcal{E} = |\phi(c') - \phi(c)| = O\left(d\,(d')^{1-\frac{k+1}{m}}\right), \tag{9}$$

   *where $\phi$ stands for the function $f$ treating weights as variables;*

2. *If $k > m - 1$, $d$ original objects can be compressed into*

$$d' = O\left(\left(\frac{d}{\varepsilon(d)}\right)^{\frac{m}{k-m+1}}\right) \tag{10}$$

   *weighted objects, such that the error is no larger than $\varepsilon(d)$.*

An implication of this theorem is that we can keep a vanishing portion of objects with vanishing error. In fact, we can reach $d' = d^\sigma$ with any $0 < \sigma < 1$. To see this, we insert $d' = d^\sigma$ into Eq. (9) and get

$$|\phi(c') - \phi(c)| = O(d^{1+\sigma(1-\frac{k+1}{m})}). \tag{11}$$

This error bound is vanishing by choosing $k > (1+\sigma^{-1})m-1$. In Eq. (9), compressing up to order $k$ is enabled by the fact that the function is at least $C^k$-differentiable, whereas the term $m$ in the exponent is a typical signature of the curse of dimensionality. This competing effect of dimensionality and differentiability is reminiscent of common error scalings in nonparametric statistics (Wasserman, 2006).

Theorem 4 seems to imply that larger $k$ always yields tighter error bounds. However, this stops being true when a cluster of $N_{m,k}$ points becomes comparable to $d$. A more careful analysis shows that the optimal choice of $k$ is $k_{\mathrm{opt}} = \Theta(d'^{1/m})$ (Theorem 7). It follows that there exists a moment matching algorithm such that the error is at most stretched-exponentially decaying as $\exp(-\mathrm{const} \times \sqrt[m]{d})$; correspondingly, we can compress $d$ original objects into

$$d' = O\left(\log^m \frac{d}{\varepsilon(d)}\right) \tag{12}$$

weighted objects, such that the error is no larger than $\varepsilon(d)$. To better understand the $d'$ lower bounds, we consider e.g., a power-law error $\varepsilon(d) = d^{-\alpha}$. Then Eq. (10) becomes $d^{-(1+\alpha)m/(k-m+1)}$, which can always be made smaller than $d$ by choosing large enough $k$. Eq. (12) becomes $((1+\alpha)\log d)^m$, which is still of order $\log^m d$. Despite we proved that compressing to $\log^m(d)$ is possible, it should be noted that when $k = k_{\mathrm{opt}}$, $N_{m,k}$ becomes comparable to $d$, so compression becomes much more computationally expensive.

*Compression rate lower bound.* In Appendix B, we prove that there exists a $d$-object distribution that cannot be compressed into less than $\propto \log^m d$ objects without inducing finite error. Hence, the compression rate $d \to \log^m d$ is *optimal* up to a constant factor, and achievable by the moment-matching method we established.

To briefly summarize the reasoning, in Theorem 8, we show that there is a $d$-point uniform distribution such that, for any $d'$-point weighted distribution, we can find a symmetric function such that its values on the two distributions differ by at least $Ad\exp[-Bd'^{1/m}]$, where $A$ and $B$ are constants. The $d$-point distribution is chosen to be "quasi-uniform" (see Definition 3). This property makes it intuitively hard to compress, since if otherwise, a big cluster of objects concentrate, we can reduce them to fewer objects by moment matching. The adversarial symmetric function is constructed as a degree $\sim d'^{1/m}$ polynomial. If we want a compression to be universally lossless, $Ad\exp[-Bd'^{1/m}]$ must be vanishing as $d \to \infty$. Solving this inequality gives the desired lower bound for $d'$, which is $\Omega(\log^m d)$.

**Numerical simulation.** Figure 2 presents a direct numerical test of the compression error scaling predicted in Theorem 4. We study the following function:

$$f(w_1, \ldots, w_d) = \frac{1}{d}\sum_{i=1}^{d}\frac{1}{10}\sum_{a=1}^{10}\mathrm{sigmoid}(\langle w_i, x_a\rangle), \tag{13}$$

where all $w_i$ and $x_a$ are $m$-dimensional vectors, and $\langle\cdot,\cdot\rangle$ denotes the inner product. We compress a fixed fraction of the original dataset across different dimensions and with varying moment-matching

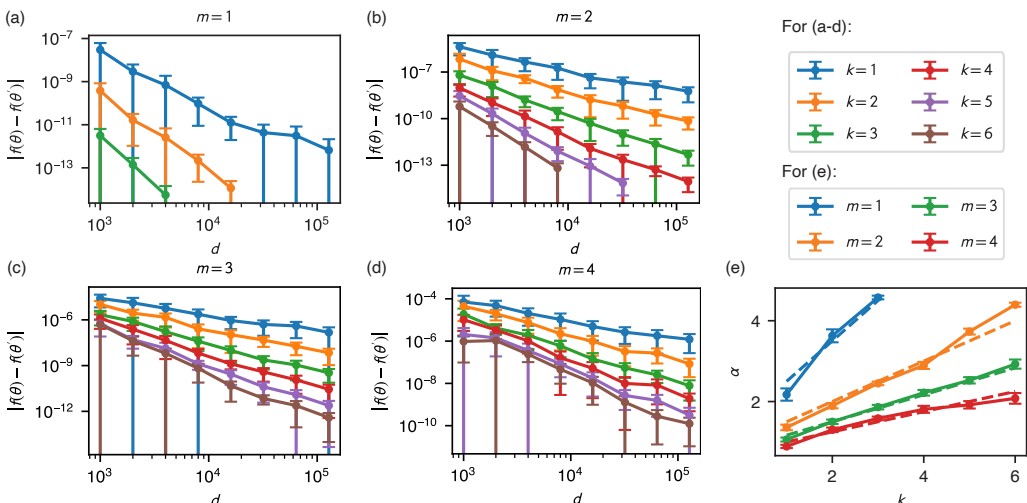

Figure 2: Error scaling for compressing a general symmetric function (Eq. (13)) using the moment-matching method. (a–d): each point shows the error in $f$ after compressing $d \to \max([0.1d], N_{m,k})$ input objects. Matching higher-order moments leads to faster error decay. (e): $\alpha$ is the fitted exponent in $|f(\theta) - f(\theta')| \propto d^{-\alpha}$. The dashed lines indicate $(k+1)/m + 0.5$, which show good agreement with the numerical results.

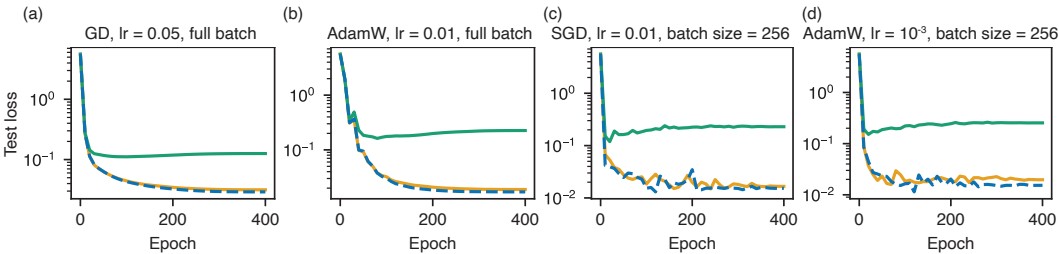

Figure 3: Compression of the training dataset in a teacher–student setup. Green dashed line: training with the original dataset of size $d = 10^4$; Orange line: training with a compressed dataset of size $10^3$, using order 5 moment matching. Blue line: training with a size-$10^3$ subset of the original dataset. Each run uses a cosine annealing learning-rate scheduler, annealing from the value shown in the plot titles to 0. Test MSE loss values are plotted every 10 epochs. It is observed that learning with the compressed dataset closely approximates the original dataset, whereas learning with a naively subsampled dataset does not.

orders. Specifically, we randomly sample vectors $x_a$ and inputs $\{w_i\}_{i \in [d]}$, perform compression, and average results over 20 trials to obtain each data point in Fig. 2 (a–d); error bars indicate standard deviation. Figure 2(e) shows that the error decays approximately as $\mathcal{E} \sim d^{-(k+1)/m-0.5}$. While Eq. (11) predicts an upper bound $\mathcal{E} \sim d^{-(k+1)/m+2}$, the observed error lies well below this bound and shows a dependency on $k$ and $m$ that is similar to the theoretical bound.

Figure 3 shows the effectiveness of dataset compression in a standard neural network training task. We consider a supervised learning problem in a teacher–student setup. Specifically, the teacher $f(x_1, x_2)$ is implemented as a random two-layer MLP of width 50 with ReLU activation. The student receives $d$ data points of the form $(x_1, x_2, \mathcal{N}(f(x_1, x_2), 3^2))$, where $x_1$ and $x_2$ are uniformly sampled from $[-1, 1]$, and the goal is to learn $f(x_1, x_2)$ using an identical network architecture. The loss function is exactly symmetric in the dataset. For full-batch update rules, which depend on the gradient of the full-batch loss, any model prediction or performance metric is thus a symmetric function of the dataset. Consequently, if the student is given a compressed dataset of size $d'$ derived from an original dataset of size $d$, the performance approximates that of training on the full dataset, and typically surpasses that of training on $d'$ naively subsampled data points (Fig. 3(a,b)).

In practice, however, stochastic update rules based on mini-batches are more common. In this case, permutation symmetry holds only in an averaged sense. Nonetheless, we find that compression remains useful. Figures 3(c,d) show the loss evolution with batch size 256, supporting this claim. Additional details are provided in Appendix E.

## 5 DYNAMICAL LOTTERY TICKET HYPOTHESIS

The lottery ticket hypothesis (Frankle & Carbin, 2018) postulates that within any sufficiently wide neural network, one can find a subnetwork that, when trained in isolation, achieves performance comparable to the original. While widely observed, its theoretical understanding has remained elusive. This section proves a corollary of our compression theory: any layer of a neural network with width $d$ can be asymptotically compressed, losslessly, to a size polylogarithmic in $d$ such that the training dynamics before and after compression are identical. We refer to this statement as the *dynamical LTH*, which can be viewed as a stronger and more quantitative variant of the original hypothesis, which only requires the final result to be identical.

The key idea behind the dynamical LTH is that predictions and loss functions are not only symmetric functions of the trained parameters, but can also be regarded as symmetric functions of the initial parameters. This holds because common update rules are *equivariant*, i.e., they commute with permutations (see Appendix C). Let $f$ denote a symmetric function, let $\mathcal{T}$ denote the training dynamics (a mapping from initial parameters to trained parameters), and let $\theta_0$ denote the initial network parameters. By equivariance, $f(\mathcal{T}(\theta_0))$ is again a symmetric function of $\theta_0$. Therefore, assuming that $f \circ \mathcal{T}$ admits a finite radius of convergence, the moment-matching compression established earlier applies directly to $f \circ \mathcal{T}$, leading to the dynamical LTH.

To make the dynamical LTH practically applicable, we must define a compressed dynamics $\mathcal{T}'$, which maps compressed initial parameters to compressed trained parameters. The formal definition is given in Appendix C, but in practice it is straightforward. For gradient-based update rules such as SGD or Adam, the compressed dynamics acting on $\theta' = \{(c_j, w_j)\}_{j \in [d']}$ is identical to the original dynamics on $\theta$, except that each gradient $\partial L / \partial w_j$ is replaced by $c_j^{-1} \partial L / \partial w_j$. With the notions of equivariance and compressed dynamics, one can prove the dynamical LTH:

**Theorem 5** (Dynamical lottery ticket hypothesis). *Let $\theta = \{w_i\}_{i \in [d]}$ be a set of permutation-symmetric trainable parameters of a neural network, and each $\|w_i\| \le R$. Suppose the model prediction $f : V^d \to \mathbb{R}$ is permutation invariant, and the training dynamics $\mathcal{T} : V^d \to V^d$ is equivariant. Also, suppose $f \circ \mathcal{T}$ satisfies Assumption 1. Then, for any initial parameter $\theta_0$, there exists a compressed weighted parameter $\theta'_0$ (which does not depend on $f$ or $\mathcal{T}$), supported on $d' = O(\log^m \frac{d}{\varepsilon(d)})$ points, such that*

$$\left| f(\mathcal{T}'(\theta'_0)) - f(\mathcal{T}(\theta_0)) \right| = \varepsilon(d). \tag{14}$$

More specific error scaling of the dynamical LTH takes the same form as in Theorems 4 and 7. Here, Assumption 1 is needed for the $\propto d$ scaling in Theorem 3 to hold. Theorem 5 not only applies to two-layer MLPs, but also naturally extends to e.g., multi-head attention. However, it requires some scrutiny to be extended to deep networks. From a similar reasoning we expect that each layer can be compressed to a vanishing portion of its original width while respecting the dynamical LTH, but the scaling of $d'$ is an open question. Regarding Assumption 1, it is theoretically not clear whether $f \circ \mathcal{T}$ respects this assumption even if $f$ does. We expect that when the single-step mappings are analytic and the number of steps is finite (compared to $d$), the analyticity of $f \circ \mathcal{T}$ follows $f$ and thus Theorem 5 holds.

A subtlety of Theorem 5 is that compression produces a "weighted neural network." However, such networks can be reduced to equivalent standard networks. To see this, recall the expression for the output in Eq. (3). For a weighted network, this becomes

$$f(x) = \sum_{j=1}^{d'} c_j v_j \, \sigma(w_j^T x). \tag{15}$$

If we redefine the second-layer weights as $v_j \leftarrow c_j v_j$, then the expression reduces exactly to the output of a standard neural network (by incorporating $c$ into the outgoing weight).

**Numerical simulation.** Figure 4 directly validates the dynamical LTH. The task is to learn the function $f(x_1, x_2)$ shown in Fig. 4(a) from $10^5$ data points of the form $(x_1, x_2, \mathcal{N}(f(x_1, x_2), 0.2^2))$, with $x_{1/2}$ sampled uniformly from $[-1, 1]$. Across a wide range of update rules and learning rates, the predictions of a wide network and its compressed counterpart are shown to be nearly indistinguishable throughout the training dynamics.

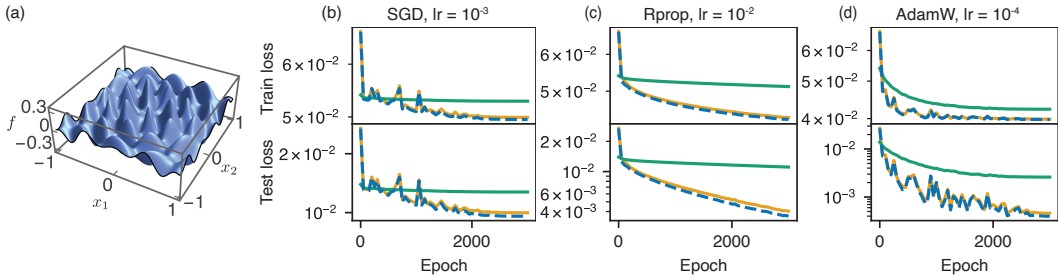

Figure 4: Dynamical LTH (Theorem 5). The demonstrated task is learning a bivariate function from noisy training data. (a) Ground-truth function $f(x_1, x_2) = J_6(20r)\cos(6\theta)$, where $r^2 = x_1^2 + x_2^2$ and $\theta = \arctan(x_2/x_1)$, known as a cylindrical harmonic. (b–d) MSE loss vs epoch under three different update rules. Green dashed line: randomly initialized network of width $10^4$; Orange line: compressed network of width $10^3$, using $k = 5$ moment matching; Blue line: random subnetwork of the $10^4$-width network, also of width $10^3$. Loss values are plotted every 50 epochs. All runs employ a cosine annealing learning-rate scheduler. Batch size is 512 for all cases, and for the three curves in each figure, we enforce identical trajectories of mini-batch choices.

## 6 IMPROVING NEURAL SCALING LAWS

Theorem 4 shows that the predictions obtained from a large number of objects can be faithfully reproduced by a much smaller set of weighted objects. This observation can be leveraged to asymptotically improve a neural scaling law. A commonly used empirical form of neural scaling laws is (Henighan et al., 2020)

$$L(d) = L_0 + (d_0/d)^\alpha, \tag{16}$$

where $L$ denotes the loss and $d$ may represent dataset size or number of parameters, or similar resources. Theorem 4 improves power-law scaling in dataset size or model parameters to at most stretched-exponential scaling. To see this, suppose we compress $d$ objects into $d' = d^\sigma$ objects, where $0 < \sigma < 1$. By Eq. (11), the loss takes the form $L(d) = L_0 + (d_0/d)^\alpha + O(d^{1+\sigma(1-\frac{k+1}{m})})$. Choosing $k$ sufficiently large ensures that the compression-induced error vanishes compared to $O(d^{-\alpha})$, leading to a improved scaling:

$$L(d') = L_0 + (d'_0/d')^{\alpha/\sigma}. \tag{17}$$

In the same spirit, we can compress $d$ to $d' = O(\log^m d)$ (inducing arbitrarily fast power law error, which is negligible) so that any power-law scaling is improved to a stretched exponential scaling as

$$L(d') = L_0 + d^{-\alpha} = L_0 + \exp(-\alpha' \sqrt[m]{d'}). \tag{18}$$

**Numerical simulation.** We demonstrate improvements in scaling laws with respect to dataset size in Fig. 5(a) and with respect to network width in Fig. 5(b). The learning tasks in Figs. 5(a) and 5(b) are the same as those in Figs. 3 and 4, respectively; additional details are provided in Appendix E. In both cases, compressing $d$ objects to $\lceil 16\sqrt{d} \rceil$ objects effectively doubles the scaling exponent. As a practical remark, the error bound in Eq. (11) guarantees sufficiently small compression error if only $k \gtrsim 13$. However, in Fig. 5 we match only up to the 6th moment and still observe a quadratic speedup. Together with Fig. 2, this suggests that in practice the error scaling can be substantially faster than the worst-case upper bound.

## 7 DISCUSSION AND OUTLOOK

In this work, we established a rigorous framework showing that any permutation-symmetric function admits strong asymptotic compression. A key advantage of the theory is that it is system-agnostic, meaning that it can be applied to any architecture or loss function. The theory has strong implications for deep learning and AI research. We showed that this result can be applied to understand the compressibility of both neural networks and datasets, a connection that has not been identified previously. This leads to a unified theory of compression for deep learning.

The central contribution of our theory is a proof of concept that it is theoretically possible to strongly compress neural networks and datasets, enabling far more efficient use of data and parameters. An

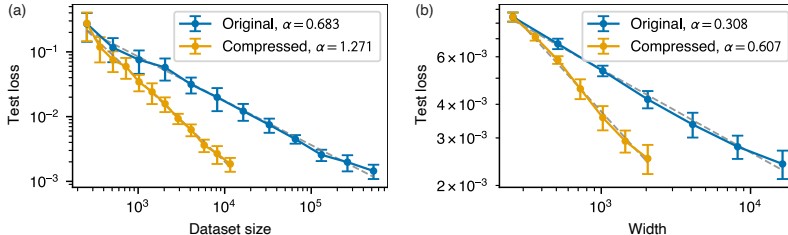

Figure 5: Improving neural scaling laws through compression. (a) MSE loss of the teacher–student task after training on an original dataset of size $d$ vs a compressed dataset of size $d'$. (b) MSE loss of the cylindrical harmonic task after training a two-layer neural network of width $d$ versus its compressed counterpart of width $d'$. In both panels, we compress $d$ objects to $d' = \lceil 16\sqrt{d}\rceil$ using $k = 6$ moment matching. The exponent $\alpha$ is obtained by fitting $L \propto d^{-\alpha}$ or $d'^{-\alpha}$.

important future direction is therefore to develop practical compression algorithms that can improve neural scaling laws at scale. We proposed a polynomial-time algorithm that achieves these scalings exactly, but it is currently too slow and memory-intensive in high dimensions. We expect future work to either optimize this algorithm or design scalable approximations. A potential limitation of moment-matching compression is slow error decay when the constituent dimension $m$ is large, reflecting the familiar curse of dimensionality. Another possible limitation is that when $m$ is large the outputs are likely to degrade in smoothness. However, many ostensibly high-dimensional objects actually lie near low-dimensional manifolds (Abbas et al., 2021). In particular, language data appear to have an effective dimension ~ 10 (Gromov et al., 2023; Du & Tanaka-Ishii, 2025). These observations suggest that our compression framework can be extended to handle high-dimensional yet structured data, where exploiting low-dimensional embeddings may greatly mitigate the apparent limitation.

Our framework also suggests new directions beyond direct compression. In particular, it points to improved data sampling strategies and model initialization schemes. For example, our proof of the dynamical LTH can be interpreted as showing that a sufficiently well-initialized network may match the performance of a randomly initialized network orders of magnitude larger. The future direction is to construct initializations (or datasets) that behave as if they were already compressed. Intuitively, well-chosen objects should be weighted and roughly equidistant, making further compression difficult. Such strategies may be connected to importance sampling (Hammersley, 2013; Bengio & Senecal, 2008) and orthogonal initialization (Saxe et al., 2014). On the theory side, many questions remain open. Extensions of our results could refine approximation rates in Barron space (Barron, 1994; Ma et al., 2022; Yang & Zhou, 2025). Finally, while our framework is rooted in the symmetric group, generalizations to other group structures may offer further insights.

## ACKNOWLEDGMENT

The authors thank Yizhou Xu for discussion. ILC acknowledges support in part from the Institute for Artificial Intelligence and Fundamental Interactions (IAIFI) through NSF Grant No. PHY-2019786. This work was also supported by the Center for Brains, Minds and Machines (CBMM), funded by NSF STC award CCF-1231216.

## REPRODUCIBILITY STATEMENT

A realization of our compression algorithm and code for generating all the figures are available at `https://github.com/WHY-David/moment_compression.git`.

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

# A  LEMMAS AND PROOFS OF THEOREMS

## A.1  FTSP

The following theorem is well known. We quote it and will use it in the proof of Theorem 1.

**Theorem 6** (Fundamental theorem of symmetric polynomials)**.** *For any symmetric polynomial* $f(x_1, \ldots, x_d)$, *where* $x_i \in \mathbb{R}$ *for all* $i \in [d]$, *there is a unique polynomial* $P$ *such that*

$$f(x_1, \ldots, x_d) = P(p_1, \ldots, p_d), \tag{19}$$

*where* $p_k$ *(* $k \in [d]$ *) is called the kth moment and is defined as*

$$p_k = \frac{1}{d} \sum_{i=1}^{d} x_i^k. \tag{20}$$

An arguably more common statement of the FTSP is that any symmetric function is a function of power sums $\sum_{i=1}^{d} x_i^k$ for $k = 0, 1, \ldots, d$, which differs from our statement by a normalization of $d$. However, our convention fixes each moment $p_k$ to constant order of magnitude. With our convention, we can treat $P$ as dependent on $d$.

## A.2  THEOREM 1

*Proof.* We write the coordinates of $w_i$ as $(w_{i,1}, \ldots, w_{i,m})$. For convenience, we use the multi-index notation $\alpha = (a_1, \ldots, a_m)$, such that $w_i^\alpha \equiv w_{i,1}^{a_1} \ldots w_{i,m}^{a_m}$. For a multi-index $\alpha$, $|\alpha| \equiv a_1 + \cdots + a_m$. Also, we define $q_\alpha = \sum_{i=1}^{d} w_i^\alpha$ (we use $q$ to distinguish from $p$ with a $1/d$ factor). Our proof is divided into a few steps: (1) Represent $f$ in terms of $q_\alpha$'s (2) Repack $\{q_\alpha\}$ into tensor moments $\{p_k\}$ (3) Show that only $p_k$ for $k = 1, \ldots, d$ are independent (4) Show that the representation in terms of $\{p_k\}$ is unique.

Generally, a polynomial $f$ is linearly spanned by monomials in the form $w_{i_1}^{\alpha_1} w_{i_2}^{\alpha_2} \ldots w_{i_N}^{\alpha_N}$ (note that $N < d$), where all subscripts are distinct and each $|\alpha_j| > 0$. Imposing permutation symmetry $S_d$, such monomials differing by index permutation must appear with the same coefficient. That is, the space of symmetric polynomials is linearly spanned by terms in the form

$$\sum_{\sigma \in S_d} w_{\sigma(1)}^{\alpha_1} w_{\sigma(2)}^{\alpha_2} \ldots w_{\sigma(N)}^{\alpha_N} \propto \sum_{\mathcal{X}(i_1, \ldots, i_N)} w_{i_1}^{\alpha_1} w_{i_2}^{\alpha_2} \ldots w_{i_N}^{\alpha_N} \tag{21}$$

Here, we introduced the notation $\mathcal{X}(i_1, \ldots, i_N)$ to denote the set of index list $(i_1, \ldots, i_N)$ such that all $i_j$ are distinct and range from 1 to $d$.

We prove by induction on $N$ that such terms can all be written as polynomials of $q_\alpha$'s. When $N = 1$, Eq. (21) is simply $q_\alpha$ by definition:

$$\sum_{i=1}^{d} w_i^\alpha = q_\alpha. \tag{22}$$

Now, suppose Eq. (21) is proven to be a polynomial of $q_\alpha$'s. Replacing $N \to N + 1$, we look at

$$\sum_{\mathcal{X}(i_1, \ldots, i_{N+1})} w_{i_1}^{\alpha_1} w_{i_2}^{\alpha_2} \ldots w_{i_{N+1}}^{\alpha_{N+1}}. \tag{23}$$

We use the set decomposition

$$\mathcal{X}(i_1, \ldots, i_N) \times \{i_{N+1}\}_1^d = \mathcal{X}(i_1, \ldots, i_{N+1}) \sqcup \mathcal{X}(i_1, \ldots, i_N) \times \{i_1\}$$
$$\sqcup \cdots \sqcup \mathcal{X}(i_1, \ldots, i_N) \times \{i_N\} \tag{24}$$

to rewrite Eq. (23) as

$$\sum_{\substack{\mathcal{X}(i_1, \ldots, i_{N+1})}} w_{i_1}^{\alpha_1} w_{i_2}^{\alpha_2} \ldots w_{i_{N+1}}^{\alpha_{N+1}} = \sum_{\substack{\mathcal{X}(i_1, \ldots, i_N) \\ i_{N+1}}} w_{i_1}^{\alpha_1} w_{i_2}^{\alpha_2} \ldots w_{i_{N+1}}^{\alpha_{N+1}}$$

$$- \sum_{\mathcal{X}(i_1, \ldots, i_N)} w_{i_1}^{\alpha_1} w_{i_2}^{\alpha_2} \ldots w_{i_N}^{\alpha_N} w_{i_1}^{\alpha_{N+1}} - \cdots - \sum_{\mathcal{X}(i_1, \ldots, i_N)} w_{i_1}^{\alpha_1} w_{i_2}^{\alpha_2} \ldots w_{i_N}^{\alpha_N} w_{i_N}^{\alpha_{N+1}}. \tag{25}$$

On the right-hand side, the first term can be written as

$$\left( \sum_{\mathcal{X}(i_1,\ldots,i_N)} w_{i_1}^{\alpha_1} w_{i_2}^{\alpha_2} \ldots w_{i_N}^{\alpha_N} \right) q_{\alpha_{N+1}}, \tag{26}$$

in which the parenthesis is a polynomial of $q_\alpha$'s by induction assumption. The second term can be written as

$$\sum_{\mathcal{X}(i_1,\ldots,i_N)} w_{i_1}^{\alpha_1+\alpha_{N+1}} w_{i_2}^{\alpha_2} \ldots w_{i_N}^{\alpha_N}, \tag{27}$$

which is a polynomial of $q_\alpha$'s as well, and so are the other terms in the ellipse. Therefore, we proved that all terms in the form of Eq. (21) can be written as polynomials of $q_\alpha$'s.

Next, we repack $q_\alpha$ into tensors. Define $p_\alpha = q_\alpha/d$. Note that the set $\{p_\alpha \mid |\alpha| = k\}$ is exactly the set of coordinates of the tensor $p_k = \sum_i w_i^{\otimes k}/d$. So indeed, the value of $f$ relies only on the tensor moments $p_k$.

To show that $f$ only requires the first $d$ moments, we try to represent any $p_k$ ($k > d$) as a function of $p_1,\ldots,p_d$. $p_k$ is a symmetric tensor of rank $k$. It is easy to see that the space of symmetric $k$-tensors is isomorphic to the space of homogeneous polynomials of degree $k$ (the argument is denoted as $u \in \mathbb{R}^m$), by the homomorphism

$$T_{i_1,\ldots,i_k} \to \sum_{i_1,\ldots,i_k} T_{i_1,\ldots,i_k} u_{i_1} \ldots u_{i_k}. \tag{28}$$

Specifically, $p_k$ is mapped to

$$s_k(u) = \frac{1}{d} \sum_{i=1}^{N} (u^\top w_i)^k \tag{29}$$

Using Theorem 6 for the variables $\{u^\top w_i \mid i = 1, 2, \ldots, d\}$, there is a polynomial $P$ such that

$$s_k(u) = P(s_1(u),\ldots,s_d(u)). \tag{30}$$

Because this holds for arbitrary $u$, it follows that $p_k$ is a function of $p_1,\ldots,p_d$ as well.

The fact that the representation of $f$ in terms of moments is unique is obvious: assume two polynomials $f$ and $f'$ are mapped to the same function. Taking the difference of the two equations $f(\theta) = P(\{p_k\})$ and $f'(\theta) = P(\{p_k\})$, we find $f(\theta) - f'(\theta) = 0$ for any $\theta$. □

### A.3 THEOREM 3

*Proof.* Choose an $m$-dimensional ball of diameter $r$ that contains all $w_i$'s, and let the center be $w_0 \in \mathbb{R}^m$. We first Taylor-expand $f(\theta)$ around $\theta_0 = (w_0,\ldots,w_0)$ up to the $k$th order. The expansion is a symmetric polynomial of degree $k$, so it can be written as a function of $(p_1,\ldots,p_k)$. By Theorem 2, we can find another set of weights $\{c'_i\}$ supported on $N_{m,k} = \binom{m+k}{k}$ points, such that

$$\sum_i c'_i (w_i - w_0)^{\otimes l} = \sum_i c_i (w_i - w_0)^{\otimes l} = p_l \quad \text{for } l = 1,\ldots,k \tag{31}$$

Let $\theta' = \{c'_i, w_i\}$, and denote its moments by $\{p'_l\}$. By construction, we have $p'_l = p_l$ for $l = 1,\ldots,k$. Since the first $k$ moments all match, there is no difference between $\phi(c)$ and $\phi(c')$ up to the $k$th order.

Next, we look at the $(k+1)$th order in the Taylor expansion of $f(\theta)$ around $\theta_0$. It is written as

$$\sum_{|\alpha|=k+1} \frac{1}{\alpha!} \partial_\alpha f(\theta_0)(\theta - \theta_0)^\alpha \tag{32}$$

It is a degree-$(k+1)$ homogeneous symmetric polynomial, and we denote it as $P_{k+1}(p_1,\ldots,p_{k+1})$. Hence,

$$f_d(\theta) - f_d(\theta') = P_{k+1}(p_1,\ldots,p_{k+1}) - P_{k+1}(p_1,\ldots,p_k,p'_{k+1}). \tag{33}$$

In $P_{k+1}(p_1,\ldots,p_{k+1})$, the only term that depends on $p_{k+1}$ is linear in itself—$\langle b_{k+1}, p_{k+1} \rangle$, where $b_{k+1}$ is a $(k+1)$-index symmetric coefficient tensor and here $\langle \cdot, \cdot \rangle$ denotes tensor contraction; all

other terms are completely determined by $\{p_l\}_{l=1}^k$. To see that $b_{k+1}$ is at most of order $d$, we use the deep-set representation: $f(\theta) = h(\sum_{i \in [d]} g(w_i))$. Then $b_{k+1}$ can be written down explicitly as

$$\langle b_{k+1}, p_{k+1} \rangle = \frac{\partial f}{\partial y} \sum_{i=1}^d \langle a_{k+1}, (w_i - w_0)^{\otimes(k+1)} \rangle$$
$$= d \langle \frac{\partial f}{\partial y} a_{k+1}, p_{k+1} \rangle, \tag{34}$$

where $y = \sum_{i \in [d]} g(w_i)$, and $\langle a_{k+1}, (w_i - w_0)^{\otimes(k+1)} \rangle$ is the $(k+1)$th order in the Taylor expansion of $g$ around $w_0$. All derivatives are taken at $\theta = \theta_0$. By our convention that $h$ and $g$ are independent of $d$, we thus have $b_{k+1} = O(d)$.

Also, since each $\|w_j - w_0\| \le r/2$ we have $p_{k+1} = O(r^{k+1})$. Therefore, we conclude that $f_d(\theta) - f_d(\theta') = O(dr^{k+1})$. □

### A.4 Sphere packing lemma

This lemma is used in proving Theorems 4 and 7.

**Lemma 1** (Sphere packing). *Given $d \gg 1$ objects in a closed $m$-dimensional ball of radius $R$: that is, $\theta = \{w_i\}_{i=1}^d$, $\|w_i\| \le R$. For any $\theta$, the diameter of the smallest ball containing $N$ points is at most of order $(N/d)^{1/m}$. That is,*

$$\sup_\theta \min_{\substack{S \subset \theta \\ |S| = N}} \operatorname{diam}(S) = R\, O\left((N/d)^{1/m}\right). \tag{35}$$

*Proof.* We use $B(x, r)$ to denote a closed $m$-ball centered at $x$.

Cover $B(0, R)$ by $M$ balls of radius $r$. By the pigeonhole principle, if $d > (N - 1)M$, some ball contains at least $N$ points, giving an $N$-point subset of diameter $\le 2r$.

We show that there exists a covering with $M = (1 + 2R/r)^m$ points. We choose a set of points $\{x_j\}_{j=1}^M \subset B(0, R)$ to form a maximal $r/2$-packing of the ball $B(0, R)$, meaning

$$\|x_i - x_j\| \ge r, \forall i \ne j, \tag{36}$$

and no further points can be added while maintaining this separation. By this definition, the balls $\{B(x_j, r)\}$ form a covering of $B(0, R)$, but the smaller balls $\{B(x_j, r/2)\}$ are mutually disjoint and contained in $B(0, R + r/2)$. By comparing volumes, we find

$$M(r/2)^m \le (R + r/2)^m. \tag{37}$$

Hence, if the radius of each ball is $r$, there exists a covering of $B(0, R)$ with $\lceil (1 + 2R/r)^m \rceil$ balls. Also requiring $d/(N-1) > \lceil (1 + 2R/r)^m \rceil$, we conclude that

$$\sup_\theta \min_{\substack{S \subset \theta \\ |S| = N}} \operatorname{diam}(S) \le \frac{4R}{\left(\frac{d}{N-1} - 1\right)^{1/m} - 1} = R\, O((N/d)^{1/m}). \tag{38}$$

□

### A.5 Theorem 4

*Proof.* Denote $N_{m,k} = \binom{m+k}{k}$. In this proof, we show that the general algorithm (Algorithm 1) with greedy clustering strategy satisfies the asserted error bounds. By the greedy strategy (also described in Appendix D), we mean that in each iteration we choose a subset $S \subseteq |\operatorname{supp}(c)|$ of $N_{m,k} + 1$ objects among the active (i.e., with nonzero weight) objects. Then we reduce one object in $S$ while maintaining up to the $k$th moment.

By Theorem 3, the output error of one step is of order $c_S r^{k+1}$, where $c_S$ is the total weight of this cluster, and $r$ is the diameter. Because the greedy strategy always look for optimizing the diameter,

after $O(d)$ steps, there is no better upper bound for $c_S$ than $d$. By Lemma 1, when there are $N$ active objects, the smallest diameter is upper-bounded by

$$O\left(\left(\frac{N_{m,k}+1}{N}\right)^{1/m}\right) = O(N^{-1/m}). \tag{39}$$

Now, we sum up the error of all the steps. Denote the function's error as $\mathcal{E} = |\phi(c') - \phi(c)|$. Then, in one step of pruning, the error is

$$\Delta\mathcal{E} = O(dr^{k+1}) = O(dN^{-(k+1)/m}). \tag{40}$$

We upper-bound the sum over $N$ by an integral using $\sum_{N=d'+1}^{d} N^{-\alpha} \le \int_{d'}^{d} N^{-\alpha} \, dN$, and find that pruning $d$ original objects into $d'$ objects results in

$$\begin{aligned}
\mathcal{E} &= O\left(\int_{d'}^{d} d\, N^{-(k+1)/m} \, dN\right) \\
&= O\left(\frac{d}{\frac{k+1}{m}-1}\left((d')^{1-\frac{k+1}{m}} - d^{1-\frac{k+1}{m}}\right)\right)
\end{aligned} \tag{41}$$

or logarithmic if $k+1 = m$; but as we are only concerned with vanishing error, we will skip the analysis of $k+1 = m$. Since $d' < d$, we conclude that the error is upper-bounded by

$$\mathcal{E} = O\left(d\,(d')^{1-\frac{k+1}{m}}\right), \tag{42}$$

which completes the proof of statement 1.

To show statement 2, we look at the equation

$$\varepsilon(d) = \mathcal{E} = O\left(d\,(d')^{1-\frac{k+1}{m}}\right) \tag{43}$$

Solving this equation with respect to $d'$ gives the asserted scaling of $d'$. $\qquad\square$

## A.6 OPTIMAL $k$ AND polylog SCALING

In establishing Theorem 7, we have been fixing $k$ as a hyperparameter which can be chosen at will. Here, we will optimize over the choice of $k$ to study the smallest achievable final size $d'$ such that the error is vanishing. In the derivation below, $k$ could scale up with $d$, but we always set $m$ to be a finite constant.

**Theorem 7.** *Assume $\|w_i\| \le R$ for all $i \in [d]$. There exists a mapping from $d$ uniformly weighted objects to $d'$ weighted objects, such that for any symmetric function $f$ satisfying Assumption 1,*

1. *The error is*

$$\mathcal{E} = |\phi_d(\theta') - \phi_d(\theta)| = O\left(d(d')^{1-1/m} \exp\left[-\frac{1}{e}(m!\rho d')^{1/m}\right]\right), \tag{44}$$

   *where $\rho$ is the radius of convergence of the function $g$ in the deep-set representation of $f$ (Eq. (4));*

2. *$d$ uniformly weighted objects can be compressed into*

$$d' = O\left(\log\frac{d}{\varepsilon(d)}\right)^m \tag{45}$$

   *weighted objects, such that the error is no larger than $\varepsilon(d)$.*

*Proof.* Essentially, we follow the same greedy strategy and the same reasoning as the proof of Theorem 4. However, since $k$ can be large, here we keep track of all factors that scales with $k$ or $d$.

Recall that the upper bound for the radius of a cluster is $((N_{m,k}+1)/N)^{1/m}$, and the upper bound for $c_S$ is $d$. Also, since the convergence radius of $g$ be $\rho$, the constant factor for the $(k+1)$th

order Taylor expansion scales as $\rho^{-k}$. Putting these together, the error of one step of pruning is upper-bounded by

$$\Delta\mathcal{E} = O\left(d\rho^{-k}\left(\frac{N_{m,k}+1}{N}\right)^{\frac{k+1}{m}}\right) \tag{46}$$

For notation simplicity, we will drop the big-$O$ notation and use $\Delta\mathcal{E}$ to refer to the upper-bound expression on the right-hand side of Eq. (46). There is an optimal $k_{\text{opt}}$ that minimizes the single-step error. We solve it by taking derivative of $\log(\Delta\mathcal{E}/d)$.

$$\begin{aligned}
\log(\Delta\mathcal{E}/d) &= -k\log\rho + \frac{k+1}{m}\left(\log(N_{m,k}+1) - \log N\right) \\
&= -k\log\rho + \frac{k+1}{m}\left(\log(m+k)! - \log k! - \log(m!N)\right) + O(1) \\
&= k\log k + \log k - \frac{k+1}{m}\log(m!N\rho) + O(1).
\end{aligned} \tag{47}$$

In the third line, we used Stirling's formula $\log(n!) = n\log n - n + \frac{1}{2}\log(2\pi n) + \frac{1}{12n} + O(n^{-3})$ when $n \to \infty$ and simplified the expression. The derivative of the above reads

$$\frac{d}{dk}\log(\Delta\mathcal{E}/d) = \log k + 1 - \frac{1}{m}\log(m!N\rho) + O(k^{-1}). \tag{48}$$

Setting the derivative to zero, we obtain

$$\log k_{\text{opt}} = \frac{1}{m}\log(m!N\rho) - 1 + O(k^{-1}) \tag{49}$$

Plugging this value into Eq. (47), we get

$$\begin{aligned}
\log(\Delta\mathcal{E}_{\text{opt}}/d) &= -k_{\text{opt}} + O(1) \\
\Rightarrow \quad \Delta\mathcal{E}_{\text{opt}} &= O\left(d\exp\left[-\frac{1}{e}(m!N\rho)^{1/m}\right]\right).
\end{aligned} \tag{50}$$

Moving forward, we integrate over $\Delta\mathcal{E}_{\text{opt}}$ from $d'$ to $d$:

$$\begin{aligned}
\mathcal{E}_{\text{opt}} &= O\left(d\int_{d'}^{d}\exp\left[-\frac{1}{e}(m!N\rho)^{1/m}\right]dN\right) \\
&= O\left(d\Gamma\left(m, \frac{1}{e}(m!\rho d')^{1/m}\right)\right),
\end{aligned} \tag{51}$$

where $\Gamma$ is the incomplete Gamma function:

$$\Gamma(m, z) \equiv \int_{z}^{\infty} t^{m-1}e^{-t}dt. \tag{52}$$

Its asymptotic behavior at $z \to \infty$ reads

$$\Gamma(m, z) = \exp\left[-z + O(z^{-1})\right]z^{m-1}\left(1 + O(z^{-1})\right). \tag{53}$$

Inserting this expansion into Eq. (51), we get the asserted error scaling in part 1 of the theorem.

To obtain part 2, we solve the inequality

$$\varepsilon(d) = \mathcal{E} = O\left(d(d')^{1-1/m}\exp\left[-\frac{1}{e}(m!\rho d')^{1/m}\right]\right) \tag{54}$$

with respect to $d'$. One can take log on both sides of this inequality and safely neglect the factor $(d')^{1-1/m}$ to eventually get

$$d' = O\left(\log\frac{d}{\varepsilon(d)}\right)^{m}. \tag{55}$$

$\square$

# B  polylog COMPRESSION RATE IS OPTIMAL

In this Appendix, we show that our algorithm of compressing $d$ objects into $O((\log d)^m)$ weighted objects is optimal up to a constant factor. The strategy to prove this is to find a $d$-point uniform distribution $\mu$, show that for any $d'$-point distribution $\mu'$ we can always find a function that evaluates to sufficiently distinct values on these two distributions.

What distribution is hard to compress? Inspired by the fact that we prioritize merging close-in-distance points in the main Algorithm 1, in an adversarial distribution, all points should be roughly equidistant from the neighbors, so there is no cluster that is particularly "easy" to compress. Hence, we introduce the following notion of quasi-uniformity:

**Definition 3** (Quasi-uniformity). *Let $D \subset \mathbb{R}^m$ be a fixed compact set with nonempty interior. A set $X_d = \{x_1, \dots, x_d\} \subset D$ is quasi-uniform if there is a constant $C_D$ (independent of $d$) such that each Voronoi cell of $x_i$ has volume less than or equal to $\frac{C_D}{d}$.*

A quasi-uniform point set obviously exists: for example, it can be a maximal $O(d^{-1/m})$-packing of the region $D$ (see the proof of Lemma 1 for the definition).

Another lemma we are going to use is the following. The basic message is that a nontrivial degree-$k$ polynomial cannot be exponentially small on most of $D$; a set of finite measure remains above $e^{-Ak}$.

**Lemma 2.** *Let $D \subset \mathbb{R}^m$ be a convex compact set, and let $|D|$ denote its volume. Let $p$ be a real polynomial on $D$ of degree $\leq k$ normalized by $\|p\|_{L^\infty(D)} \equiv \sup_{x \in D} |p(x)| = 1$. For any $t > 0$, define*

$$S_t = \{x \in D : |p(x)| \geq t\}. \tag{56}$$

*Then*

$$\frac{|S_t|}{|D|} \geq 1 - (t/2)^{1/k}. \tag{57}$$

*Proof.* We begin by quoting a multivariate Remez inequality (Theorem 1.2 in Brudnyi & Yomdin (2015)): for any measurable $E \subset D$ with $\lambda = \frac{|E|}{|D|} \in (0, 1]$ and any real polynomial $q$ of degree $\leq k$,

$$\|q\|_{L^\infty(D)} \leq T_k\left(\frac{1 + (1 - \lambda)^{1/m}}{1 - (1 - \lambda)^{1/m}}\right) \|q\|_{L^\infty(E)}, \tag{58}$$

where $T_k$ is the Chebyshev polynomial of the first kind.

For $t \in (0, 1)$, we use $E_t$ to denote the sublevel set:

$$E_t = \{x \in D : |p(x)| \leq t\}, \qquad \lambda_t = \frac{|E_t|}{|D|}. \tag{59}$$

Applying (58) with $q = p$, $E = E_t$ and using $\|p\|_{L^\infty(D)} = 1$ gives

$$1 \leq T_k\left(\frac{1 + (1 - \lambda_t)^{1/m}}{1 - (1 - \lambda_t)^{1/m}}\right) t. \tag{60}$$

Then, we make simplifications to the Chebyshev term in Eq. (60). For $x \geq 1$,

$$T_k(x) = \tfrac{1}{2}\left(x + \sqrt{x^2 - 1}\right)^k + \tfrac{1}{2}\left(x - \sqrt{x^2 - 1}\right)^k \geq \tfrac{1}{2} x^k. \tag{61}$$

Moreover, since $v \mapsto v^{1/m}$ is increasing and $v^{1/m} \geq v$ on $[0, 1]$,

$$\frac{1 + (1 - \lambda)^{1/m}}{1 - (1 - \lambda)^{1/m}} \geq \frac{1}{1 - (1 - \lambda)^{1/m}} \geq \frac{1}{\lambda} \qquad \text{for } \lambda \in (0, 1]. \tag{62}$$

Combining (61) and (62) in (60), we obtain

$$1 \leq \tfrac{1}{2} \lambda_t^{-k} t \implies \lambda_t \leq (t/2)^{1/k}. \tag{63}$$

Now we pass to the superlevel set $S_t$. Note that $|S_t| + |E_t| = |D|$. Then

$$\frac{|S_t|}{|D|} = 1 - \lambda_t \geq 1 - (t/2)^{1/k}. \tag{64}$$

$\square$

The following theorem constructs an adversarial polynomial (i.e., has moderately large error however we compress). Remember from Sec. 2 that our global assumption for functions is this paper is that they have finite convergence radius, so the polynomial constructed here lies within the assumption.

**Theorem 8.** *Let $\mu$ and $\mu'$ be non-negative distributions supported on $D$: $\mu = \sum_{i=1}^{d} \delta_{x_i}$, $\mu' = \sum_{j=1}^{d'} c_j \delta_{y_j}$ where all $x_i, y_j \in D$ and $c_j > 0$. There exists a $\mu$ such that for any $\mu'$, there exists a polynomial $g$ and constants $A, B > 0$ such that*

$$\left| \int_D g \, d\mu - \int_D g \, d\mu' \right| \geq A d \exp[-B d'^{1/m}]. \tag{65}$$

*Proof.* Let $k$ be the smallest integer with $N_{m,k} = \binom{m+k}{m} > d'$. Let $q : \mathbb{R}^m \mapsto \mathbb{R}$ be a degree-$k$ polynomial. Since there are $N_{m,k}$ parameters in $q$, there exists a non-zero polynomial such that $q(y_1) = \cdots = q(y_{d'}) = 0$. Also, $q$ is normalized so that $\|q\|_{L^\infty(D)} = 1$. Let $g = q^2$ be the adversarial function that will be shown to satisfy Eq. (65). For $g$ and $\mu'$, we have

$$\int g \, d\mu' = \sum_{j=1}^{d'} c_j q(y_j)^2 = 0. \tag{66}$$

Next, we consider $\int_D g \, d\mu$. Denote the point set of $\{x_i\}_{i=1}^{d}$ by $X_d$. Quasi-uniformity of $X_d$ implies that the number of points inside a region is comparable to the volume: $\#(X_d \cap S) \geq \frac{d}{c_D}|S|$ for some constant $c_D > 0$. Let $S$ be the superlevel set $S_t$, we have

$$\int_D g \, d\mu = \sum_{i=1}^{d} q(x_i)^2 \geq \frac{1}{c_D} d |S_t| t^2 \geq \frac{|D|}{c_D} d \left(1 - (t/2)^{1/k}\right) t^2, \tag{67}$$

where we used Lemma 2 in the last line. We further plug in $t = 2e^{-Ck}$ to have

$$\int_D g \, d\mu \geq \frac{|D|}{c_D} d \left(1 - e^{-C}\right) e^{-2Ck}. \tag{68}$$

Putting this together with $\int_D g \, d\mu' = 0$ and denote $A = \frac{|D|}{c_D} \left(1 - e^{-C}\right)$,

$$\left| \int_D g \, d\mu - \int_D g \, d\mu' \right| \geq A d e^{-2Ck}. \tag{69}$$

Finally, recall that $k$ is the minimal integer with $N_{m,k} > d'$. Since $N_{m,k} = \binom{m+k}{m} \sim k^m/m!$, there exist constants $c_1$ and $c_2$ (depending only on $m$) such that

$$c_1 d^{1/m} \leq k \leq c_2 d^{1/m}. \tag{70}$$

Therefore,

$$\left| \int_D g \, d\mu - \int_D g \, d\mu' \right| \geq A d \exp[-B d'^{1/m}], \tag{71}$$

where $B = -2C \cdot c_2$. $\square$

Finally, we show that Theorem 8 leads to a $\Theta((\log d)^m)$ compression lower bound. We require the compression error to be at most $\varepsilon(d)$, which is a vanishing function when $d \to \infty$. This is not possible if

$$A d \exp[-B d'^{1/m}] \geq \varepsilon(d), \tag{72}$$

which is equivalent to

$$d' \leq \frac{1}{B} \left(\log \frac{A d}{\varepsilon(d)}\right)^m. \tag{73}$$

A common choice for $\varepsilon(d)$ is $\varepsilon(d) \propto d^{-\alpha}$. In this case, the right-hand side of the above inequality is proportional to $(\log d)^m$. Therefore, a universal compression from $d$ objects to $O((\log d)^m)$ is optimal.

## C   FORMAL THEORY ON THE DYNAMICAL LTH

In this Appendix, we formulate all ideas mentioned in Sec. 5 with mathematical rigor.

Let $S_d$ be the permutation group of $d$ elements. Let $V = \mathbb{R}^m$. We define the representation $R : S_d \mapsto \mathrm{End}(V^d)$ as

$$R(\sigma)(w_1, \ldots, w_d) = (w_{\sigma(1)}, \ldots, w_{\sigma(d)}). \tag{74}$$

Note that the definition of $f : V^d \mapsto V^d$ being symmetric is equivalent to: for any $\sigma \in S_d$, $f \circ R(\sigma) = f$.

**Definition 4** (Equivariant map). *A function $\mathcal{T} : V^d \mapsto V^d$ is called an equivariant map if for any $\sigma \in S_d$,*

$$\mathcal{T} \circ R(\sigma) = R(\sigma) \circ \mathcal{T}. \tag{75}$$

The dynamics induced by equivariant maps has been studied in conventional settings of dynamical systems (Field, 1980), but has not received much attention in deep learning. In fact, almost all update rules that we commonly use are equivariant, which we will verify for SGD and Adam later in this Appendix. Because compositions of equivariant maps are also equivariant, it follows that the entire training dynamics (i.e., the mapping from initial model parameters to trained parameters) is equivariant.

The following lemma shows that the composition of a symmetric function with an equivariant mapping is also a symmetric function.

**Lemma 3.** *If $f$ is a symmetric function and $\mathcal{T}$ is an equivariant map, then $f \circ \mathcal{T}$ is a symmetric function.*

*Proof.* For any $\sigma \in S_d$,

$$(f \circ \mathcal{T}) \circ R(\sigma) = (f \circ R(\sigma)) \circ \mathcal{T} = f \circ \mathcal{T}. \tag{76}$$

$\square$

Thanks to this lemma, we can treat $f \circ \mathcal{T}$ as a single symmetric function, and thus we expect that compressing the weights by our moment matching method induces a vanishing error in the value of $f \circ \mathcal{T}$, that is, literally any prediction of the trained model.

The following lemma establishes a general representation of equivariant maps in terms of moments, so it can be viewed as an extension of the FTSP.

**Lemma 4.** *$\mathcal{T}$ is an equivariant map if and only if for all $i \in [d]$,*

$$\mathcal{T}_i(\theta) = T(w_i, \boldsymbol{p}), \tag{77}$$

*where $\boldsymbol{p}$ is a collective notation for $\{p_k = \frac{1}{d} \sum w_i^{\otimes k}\}_{k=1}^{d-1}$. Note that $T$ is a function that does not depend on $i$, and is uniquely determined by $\mathcal{T}$.*

*Proof.* First, we verify that $\mathcal{T}_i(\theta) = T(w_i, \boldsymbol{p})$ for any function $T$ is an equivariant map. Note that $\sigma(\boldsymbol{p}) = \boldsymbol{p}$. Following the definition of equivariance,

$$\mathcal{T}_i \circ \sigma(\theta) = \mathcal{T}_i(w_{\sigma(1)}, \ldots, w_{\sigma(d)}) = T(w_{\sigma(i)}, \boldsymbol{p}) = \mathcal{T}_{\sigma(i)}(\theta). \tag{78}$$

Thus, $\mathcal{T} \circ \sigma = \sigma \circ \mathcal{T}$.

The rest of this proof is to show that all equivariant maps can be written in the asserted form. For a fixed $i$, let $\hat{\sigma} \in S_d$ be any permutation group element such that $\hat{\sigma}(i) = i$. Consider the $i$th component of the equation $\mathcal{T}(\hat{\sigma}(\theta)) = \hat{\sigma}(\mathcal{T}(\theta))$:

$$\mathcal{T}_i(\hat{\sigma}(\theta)) = \mathcal{T}_i(\theta). \tag{79}$$

This means that $\mathcal{T}_i$ is invariant under any permutation on $[d] \backslash \{i\}$, which forms a representation of $S_{d-1}$. By Theorem 1, $\mathcal{T}_i$ can be uniquely written as a function of $w_i$ and power sums $\sum_{j \neq i} w_j^{\otimes k}$. Since $\sum_{j \neq i} w_j^{\otimes k}$ is uniquely determined by $w_i$ and $p_k$ as $dp_k - w_i^{\otimes k}$, we conclude that there is a unique function $T_i$ such that

$$\mathcal{T}_i(\theta) = T_i(w_i, \boldsymbol{p}). \tag{80}$$

The final task is to show that $T_i$ does not depend on $i$. We use $\pi_{i,j}$ to denote the permutation group element that only exchanges $i$ and $j$. The $i$th component of the equation $\mathcal{T}(\pi_{i,j}(\theta)) = \pi_{i,j}(\mathcal{T}(\theta))$ reads

$$T_i(w_j, \boldsymbol{p}) = T_j(w_j, \boldsymbol{p}), \tag{81}$$

which completes the proof. $\qquad\square$

We use Lemma 4 to unambiguously define the compressed training dynamics:

**Definition 5** (Compressed dynamics). *Suppose a training dynamics is determined by an equivariant map $\theta = \mathcal{T}(\theta_0)$ (arbitrarily initialized weights to trained weights), which can be written as in Eq. (77). For the weighted parameters $\theta' = \{c_j, w_j\}_{j \in [d]}$ ($c_j$ never changes with the dynamics; some $c_j$ might be zero so that they are actually pruned), we define the compressed dynamics $\mathcal{T}'$ as*

$$\mathcal{T}'_j(\theta') = T(w_j, \boldsymbol{p}'), \tag{82}$$

*where $\boldsymbol{p}'$ is a collective notation for $\{p'_k = \frac{1}{d} \sum_j c_j w_j^{\otimes k}\}_{k=1}^d$. Note that $\mathcal{T}'$ is uniquely determined by $\mathcal{T}$.*

The mapping from original learning dynamics to compressed ones is in fact easy in practice. Below are some common examples.

1. Stochastic gradient descent (SGD). Consider a two-layer neural network used in supervised learning. For simplicity, we write the output as $y = \sum_{i=1}^d g(w_i, x)$, which is symmetric in $\theta = (w_1, \ldots, w_d)$. Each time we choose a batch of training data $\{(x_a, y_a)\}_{a \in \mathcal{B}}$. We denote the per-sample loss function as $\ell_a = \ell(y(\theta, x_a), y_a)$. The SGD update rule is

$$
\begin{aligned}
\mathcal{T}_i(\theta) &= w_i - \eta \mathop{\mathbb{E}}_{a \in \mathcal{B}} \frac{\partial \ell_a}{\partial w_i} \\
&= w_i - \eta \mathop{\mathbb{E}}_{a \in \mathcal{B}} \frac{\partial \ell_a}{\partial y} \frac{\partial g(w_i, x_a)}{\partial w_i}
\end{aligned}
\tag{83}
$$

where $\eta$ is the learning rate. Note that $\partial \ell_a / \partial y$ is a function of $y$, which is thus permutation invariant in $\theta$. We can explicitly compute $(\mathcal{T} \circ R(\sigma))(\theta)$ and $(R(\sigma) \circ \mathcal{T})(\theta)$, which are both equal to

$$w_{\sigma(i)} - \eta \mathop{\mathbb{E}}_{a \in \mathcal{B}} \frac{\partial \ell_a}{\partial y} \frac{\partial g(w_{\sigma(i)}, x_a)}{\partial w_{\sigma(i)}}, \tag{84}$$

so SGD is indeed equivariant.

Then, we derive the compressed SGD. Remember that the weighted neurons compute the output as $y = \sum_{j=1}^{d'} c_j g(w_i, x)$. Eq. (83) can indeed be written in the form $T(w_i, \boldsymbol{p})$ because $\partial \ell_a / \partial y$ is a function of $\boldsymbol{p}$, and $\partial g(w_i, x_a)/\partial w_i$ solely depends on $w_i$. Therefore, the compressed update rule still looks like

$$\mathcal{T}'_j(\theta) = w_j - \eta \mathop{\mathbb{E}}_{a \in \mathcal{B}} \frac{\partial l_a}{\partial y} \frac{\partial g(w_i, x_a)}{\partial w_j}. \tag{85}$$

However, we emphasize that it is not $w_j - \eta \mathbb{E}_{a \in \mathcal{B}} \partial \ell_a / \partial w_j$, because

$$\frac{\partial \ell_a}{\partial w_j} = \frac{\partial \ell_a}{\partial y} c_j \frac{\partial g(w_j, x_a)}{\partial w_j} \tag{86}$$

Effectively, whenever there is a gradient $\partial L / \partial w_j$ in the original update, we should replace it by $c_j^{-1} \partial L / \partial w_j$. This rule applies for all other gradient-based updates as well.

Finally, we remark on non-deterministic updates. It seems that choosing mini-batches turns the update into a stochastic process, which complicates the problem. But in fact, for any fixed trajectory of mini-batch choices, the update is explicitly equivariant (choosing a mini-batch breaks the permutation symmetry among the training dataset, but not the permutation symmetry of neurons). Therefore, we always expect to see good agreement between the original and compressed dynamics if we use the same choice of mini-batches for both.

2. Adam. The Adam update rule for standard (i.e., uniformly weighted) parameters reads

$$
\begin{aligned}
g_t &\leftarrow \nabla_\theta \mathop{\mathbb{E}}_{a \in \mathcal{B}} \frac{\partial \ell_a}{\partial \theta}(\theta_{t-1}) \\
m_t &\leftarrow \beta_1 m_{t-1} + (1 - \beta_1) g_t \\
v_t &\leftarrow \beta_2 v_{t-1} + (1 - \beta_2) g_t^2 \\
\hat{m}_t &\leftarrow \frac{m_t}{1 - \beta_1^t} \\
\hat{v}_t &\leftarrow \frac{v_t}{1 - \beta_2^t} \\
\theta_t &\leftarrow \theta_{t-1} - \eta \frac{\hat{m}_t}{\sqrt{\hat{v}_t} + \epsilon},
\end{aligned} \tag{87}
$$

where $t$ denotes time step. To check that Adam is equivariant, we only need to note that the gradient ($g_t$) for $w_i$ takes the form

$$
\mathop{\mathbb{E}}_{a \in \mathcal{B}} \frac{\partial \ell_a}{\partial y} \frac{\partial g(w_i, x_a)}{\partial w_i}, \tag{88}
$$

where $\partial \ell_a / \partial y$ is symmetric, and $\partial g(w_i, x_a)/\partial w_i$ is solely a function of $w_i$. Using this fact, it is straightforward to check that $(\mathcal{T} \circ R(\sigma))(\theta)$ and $(R(\sigma) \circ \mathcal{T})(\theta)$ are identical.

To define the compressed version of Adam, we need to keep the gradient exactly as in Eq. (88), as in our discussion on SGD. This in turn tells us that we just need to replace $\partial L / \partial w_j$ by $c_j^{-1} \partial L / \partial w_j$. Special to Adam, because $1/c_j$ appears in both $\hat{m}_t$ and $\sqrt{\hat{v}_t}$, the compressed update rule is basically the same as the original if we neglect the small $\epsilon$. Indeed, in the numerical simulations we conducted with AdamW (the reasoning is the same as Adam), we did not observe any visible difference whether or not to scale the gradients by $1/c_j$.

Finally, we prove the dynamical LTH, which we restate here.

**Theorem** (Theorem 5, Dynamical lottery ticket hypothesis). *Let $\theta = \{w_i\}_{i \in [d]}$ be a set of permutation-symmetric trainable parameters of a neural network, and each $\|w_i\| \le R$. Suppose the model prediction $f : V^d \to \mathbb{R}$ is permutation invariant, and the training dynamics $\mathcal{T} : V^d \to V^d$ is equivariant. Also, suppose $f \circ \mathcal{T}$ satisfies Assumption 1. Then, for any initial parameter $\theta_0$, there exists a compressed weighted parameter $\theta_0'$ (which does not depend on $f$ or $\mathcal{T}$), supported on $d' = O(\log^m \frac{d}{\varepsilon(d)})$ points, such that*

$$
\left| f(\mathcal{T}'(\theta_0')) - f(\mathcal{T}(\theta_0)) \right| = \varepsilon(d). \tag{89}
$$

*Proof.* We compress $\theta_0$ using moment matching. By construction, for all $l = 0, 1, \ldots, k$ we have

$$
\sum_{i=1}^d (w_0)_i^{\otimes l} = \sum_{j \in S} c_j (w_0)_j^{\otimes l} \tag{90}
$$

For any $f$, $f \circ \mathcal{T}$ and $f \circ \mathcal{T}'$ can both be written as a function of moments. In this representation, using the definition in Eq. (82), one can check that they are exactly the same function. The difference is that $f \circ \mathcal{T}$ takes in $(p_0)_k = \frac{1}{d} \sum_i (w_0)_i^{\otimes k}$, while $f \circ \mathcal{T}'$ takes in $(p_0')_k = \frac{1}{d} \sum_j c_j (w_0)_j^{\otimes k}$; they are identical in the first $k$ moments by construction. Using Theorem 4, we conclude that using large enough $k$, the difference between $f(\mathcal{T}(\theta_0))$ and $f(\mathcal{T}'(\theta_0'))$ can always be made vanishing, with the same error upper-bound as asserted in Theorem 4. Ultimately, using the optimal $k_{\text{opt}}$ given in Theorem 7, we achieve $d' = O\left(\log^m \frac{d}{\varepsilon(d)}\right)$ with error at most $\varepsilon(d)$. $\square$

## D   AN EXAMPLE OF THE MOMENT-MATCHING COMPRESSION ALGORITHM

Here, we describe the concrete algorithm for compression that is used in all our numerical simulations.

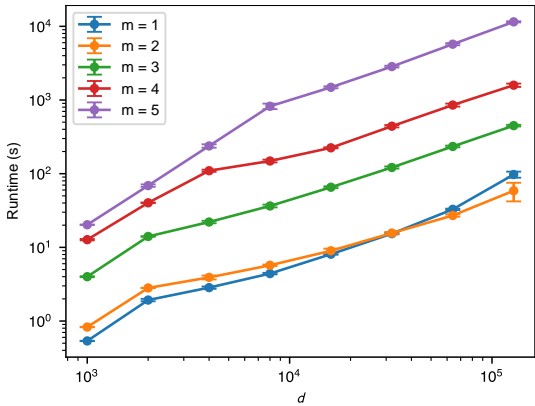

Figure 6: Runtime benchmark of the hybrid compression algorithm. See description in Sec. D. The original dataset is i.i.d. uniformly sampled in an $m$-dimensional cube. The moment-matching order is $k = 5$ for all trials in this plot. Each marker on the plot is an average over 5 trials, with error bar representing one standard deviation.

Recall that in Algorithm 1 we identified two main steps of the algorithm: (1) clustering and (2) moment matching. We first describe our moment matching strategy. For moment-matching, we consistently use Algorithm 2, in which $\text{vect}(\cdot)$ represents flattening all the moments into a vector. We remark that the existence of Algorithm 2 is effectively a constructive proof of Tchakaloff's theorem 2.

Our actual clustering strategy is slightly more involved, because finding the smallest cluster in $d \gg 1$ points is known to be NP hard. We implement a coarser $k$-means clustering instead. Only when $|\text{supp}(c)|$ becomes close to the desired stopping size $d'$, we switch to a greedy strategy, since the diameters of clusters are likely to be large when $|\text{supp}(c)|$ is small. Concretely, in a $k$-means round, we divide $\theta$ into $\propto |\text{supp}(c)|/N_{m,k}$ clusters. Then we apply Algorithm 2 to each cluster containing more than $N_{m,k}$ objects in parallel. In a greedy round, we find the approximately smallest cluster of $N_{m,k} + 1$ points, which is implemented using the $k$-nearest neighbor algorithm provided by the `faiss` package (Douze et al., 2024).

For the above $k$-means/greedy hybrid strategy, we present the runtime benchmark in Fig. 6. The calculation is conducted on a personal computer with Apple M4 Pro CPU. The runtime is observed to be roughly proportional to $d$. This is because the number of iterations over moment-matching reduction of support is proportional to $d$.

Finally, we remind the reader that our error bound Theorems 4 and 7 are proved for the greedy strategy, where in each round one finds the smallest cluster of $N_{m,k} + 1$ objects and reduce one object. For the above-mentioned hybrid clustering strategy, there is no theoretical guarantee as strong as Theorem 4 for the error, but all numerical simulation turns out to meet our expectation.

---

**Algorithm 2:** Reducing support while matching moments

---

**Input** : Moment matching order $k$

**Input** : $N$ weighted parameters $\{(c_j, w_j)\}_{j=1}^N$, where $N_{m,k} = \binom{m+k}{k}$

**Output:** Adjusted weights $\{c_j\}$ where $|\text{supp}(c)| \le N_{m,k}$

**function** $\phi(w) = \text{vect}(1, w, w^{\otimes 2}, \ldots, w^{\otimes k})^\top$    /* $\dim \phi(w) = N_{m,k}$       */;

**while** $|\text{supp}(c)| > N_{m,k}$ **do**

    Let $\text{supp}(c) = \{j_1, \ldots, j_{|\text{supp}(c)|}\}$; $A = \begin{pmatrix} \phi(w_{j_1}) & \phi(w_{j_2}) & \cdots & \phi(w_{j_{|\text{supp}(c)|}}) \end{pmatrix}$;

    Find a nonzero $v \in \mathbb{R}^{|\text{supp}(c)|}$ such that $Av = 0$; Ensure that at least one $v_j > 0$ ;

    $t = \min_{j \in \text{supp}(c): v_j > 0} c_j/v_j$;

    $c_j \leftarrow c_j - tv_j$.

**end**

---

# E DETAILS ON NUMERICAL EXPERIMENTS

We studied two main numerical tasks in Figs. 3, 4 and 5. For all these experiments, the loss function in training is the mean squared error (MSE) loss. The test loss shown in figures is the MSE loss evaluated on a randomly sampled dataset of the form $(x_1, x_2, f(x_1, x_2))$, where $f(x_1, x_2)$ is the ground truth function. For Figs. 3 and 4, the test dataset size is $10^5$, and for Fig. 5 it is $2 \times 10^4$. All unspecified training hyperparameters follow PyTorch defaults. In particular, for AdamW, they are $\beta_1 = 0.9$, $\beta_2 = 0.999$, (weight decay) $\lambda = 0.01$, and $\epsilon = 10^{-8}$.

Fig. 3 and 5(a) concerns compressing the training dataset. The task is function fitting in a teacher-student setup, described in Sec. 4.2. When the training dataset is weighted, the data loader is implemented as follows: We draw i.i.d. samples from $\{w_j\}_{j \in [d']}$, using $\{c_j\}_{j \in [d']}$ as the unnormalized probability distribution, to form a mini-batch.

Fig. 4 and 5(b) concerns compressing the width of a two-layer neural network. The task is fitting an oscillating bivariate function known as a cylindrical harmonic, described in the caption of Fig. 4. The training dataset size for Fig. 4 is $10^5$, and for Fig. 5 is $2 \times 10^4$.

In Fig. 5, we show the test MSE loss scaling with respect to training dataset size (a) and neural network width (b). For both (a) and (b), the update rule is AdamW. The learning rate is initially $0.001$, and is modulated by a cosine annealing learning rate scheduler, reaching 0 at the final epoch. Each data point is obtained by averaging 10 random instances of the train and test dataset and the neural network initialization, and the error bars indicate the standard deviation. For (a), we train each instance for 2048 epochs, each epoch containing one mini-batch of size 512, so that there is a constant compute budget for the original and the compressed datasets. For (b), we train each instance for 2000 epochs, each epoch enumerates over the train dataset. The batch size is 128.

# F PERMUTATION SYMMETRY IN ATTENTION MODULES

In principle, the compression theory developed in this work can be applied to attention mechanisms in two distinct ways. We briefly alluded to these ideas in Section 2; here, we provide a self-contained and more detailed discussion. The two applications concern $(1)$ the compression of the query and key weight matrices, and $(2)$ the compression of attention heads.

**Compression of query and key matrices.** The first application serves as a straightforward verification of the theory. Consider the query and key matrices $W_Q$ and $W_K$. The attention logits depend only on their product, which can be written as

$$a = a(W_Q W_K) \tag{91}$$

with

$$W_Q W_K = \sum_{i=1}^{d} w_Q^i \left( w_K^i \right)^T, \tag{92}$$

where $w_Q^i$ denotes the $i$-th row of $W_Q$ and $w_K^i$ the $i$-th column of $W_K$. Since the index $i$ is a dummy summation index, its ordering is immaterial. This reveals an explicit permutation symmetry among the pairs $\{(w_Q^i, w_K^i)\}_{i=1}^{d}$.

Because the output depends only on the sum of these outer products, the symmetry implies that the collection of these row–column pairs can be compressed. Moreover, if the left dimension of $W_Q$ is $m$, then $W_Q W_K$ has rank at most $m$, and the effective number of degrees of freedom is independent of $d$. Consequently, one can achieve not merely a $\mathrm{polylog}(d)$ compression but, in fact, a constant-size representation. This serves as a useful sanity check for the consistency of our general theory.

A more interesting direction arises when the key–query interaction becomes nonlinear. For instance, one may consider replacing the bilinear form with

$$\sum_{i=1}^{d} w_Q^i s\left(w_K^i, X\right)^T, \tag{93}$$

where $s$ is a nonlinear function and $X$ denotes the input data. In this setting, the permutation symmetry persists, and our theory guarantees that an $\mathrm{polylog}(d)$-size compressed representation of this nonlinear attention layer is achievable in principle.

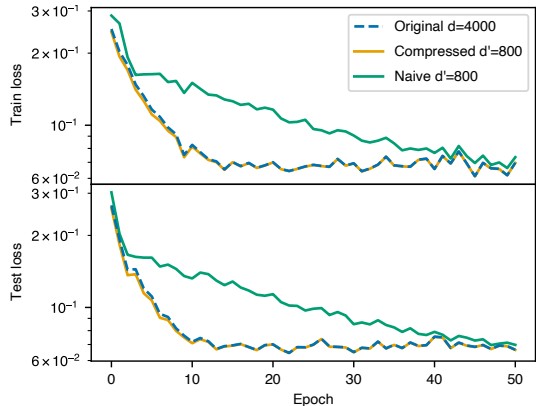

Figure 7: Dynamical LTH extended to transformers. The training dynamics of a large, $d_{\text{heads}}$ = 4000-head attention model shows good agreement with its compressed multi-head attention model with $d'_{\text{heads}}$ = 800 heads. See description of the task in Appendix F.

**Compression of attention heads.**    The second application concerns the compression of entire attention heads, which is intrinsically more meaningful. Consider an attention layer with $d$ heads, and let

$$A_i = B(w_i, X) \tag{94}$$

denote the output of the $i$th head, where $w_i$ denotes its trainable parameter (including the query/key/value weight matrices $W_{Q/K/V,i}$ in the $i$th head) and $B$ the head-level transformation. The standard output (denoted by $y$, of the attention layer is

$$y = U \operatorname{concat}(A_1, \dots, A_d), \tag{95}$$

where $U \in \mathbb{R}^{z \times dh}$ is the output projection matrix, $z$ is the dimension of the final output, and $h$ is the dimension of each head output. Partitioning $U$ into blocks $U_i \in \mathbb{R}^{z \times h}$, this expression becomes

$$y = \sum_{i=1}^{d} U_i B(w_i, X). \tag{96}$$

This summation structure again exposes a permutation symmetry: the parameters

$$\theta_i = (U_i, w_i) \tag{97}$$

enter the model only through their sum over $i$, and their ordering is irrelevant. By the general theory, any such collection of $d$ symmetric objects admits a compressed representation of size $O(\operatorname{polylog}(d))$. Hence, the entire set of attention heads can be compressed to $\operatorname{polylog}(d)$ effective parameters while preserving the functional form of the output.

With the permutation symmetry among heads, it is easy to formulate a similar dynamical LTH for multi-head attention. Figure 7 is a numerical demonstration of LTH in transformers. Here, the task is in-context learning on random piecewise-linear functions. We consider a scalar in-context regression task in which each episode defines a random continuous piecewise-linear function $f : [0, 1] \to \mathbb{R}$. For a given episode, we first draw an initial value $f_0 \sim \mathcal{N}(0, \sigma_{f_0}^2)$ and segment slopes $s_j \sim \mathcal{N}(0, \sigma_s^2)$ for $j = 0, \dots, K-1$, with $K = 16$, $\sigma_{f_0} = 0.5$ and $\sigma_s = 1.0$. The interval $[0, 1]$ is partitioned into $K$ equal sub-intervals of length $1/K$, and $f$ is defined by enforcing continuity and setting the slope on segment $j$ to $s_j$. For each episode we sample $n_{\text{ctx}} = 8$ context inputs $x_i \sim \operatorname{Unif}[0, 1]$ with noisy observations $y_i = f(x_i) + \varepsilon_i$, where $\varepsilon_i \sim \mathcal{N}(0, \sigma_{\text{noise}}^2)$ with $\sigma_{\text{noise}} = 0.3$, together with an additional query point $x_* \sim \operatorname{Unif}[0, 1]$ and a clean target $y_* = f(x_*)$. The episode is presented to the model as a scalar sequence of tokens $[x_1, y_1, \dots, x_{n_{\text{ctx}}}, y_{n_{\text{ctx}}}, x_*] \in \mathbb{R}^{2n_{\text{ctx}}+1}$ (token dimension $d_{\text{in}} = 1$), which is processed by a single-layer causal multi-head attention module with $d_{\text{heads}} = 4000$ heads and per-head dimension $d_{\text{head}} = 2$. The model outputs a scalar prediction $\hat{y}_*$ from the final token position. We compare three variants that share the same initialization: (i) the full model with all 4000 heads, (ii) a compressed model obtained by reducing the number of heads to $d'_{\text{heads}} = 800$ via compression of order $k = 3$ (the effective dimension of each symmetric object is $m = 8$), and (iii)

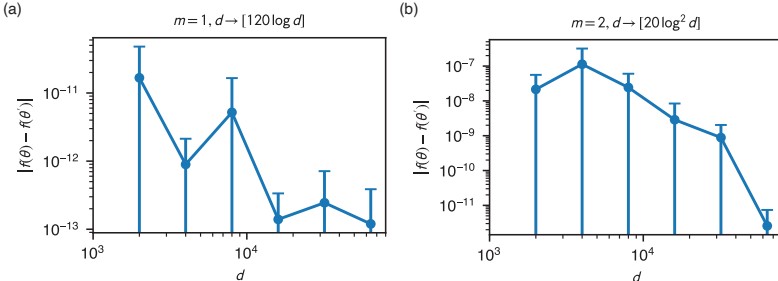

Figure 8: Error scaling of compressing a general symmetric function using the moment-matching method. Here, various different values of $k$ (the order of moment matching) are attempted, from small to large, until the smallest error is found. Each data point is an average over 5 random instances, plotting the average with error bar standing for one standard deviation.

a naive head-pruned model where 800 heads are selected uniformly at random and the remaining heads are discarded. All three models are trained with Adam (learning rate $10^{-4}$, batch size 256) on the same sequence of mini-batches, using 5 gradient steps per epoch for 50 epochs. At epoch 0 (before training) and after each epoch we report the MSE loss, for both training-like and test-like evaluations.

## G    NUMERICAL COMPRESSION TO $\mathrm{polylog}(d)$

In this section, we numerically demonstrate the possibility to the optimal rate, that is, compressing $d$ objects into $O(\log^m d)$ objects. As we argued in Theorem 7 and Appendix D, compressing to this rate is computationally heavy for the moment-matching algorithm, so we only show it in low dimensions, i.e., $m = 1$ or $2$.

The error scaling is shown in Fig. 8. The function we study is

$$f(w_1, \ldots, w_d) = \frac{1}{d} \sum_{i=1}^{d} \frac{1}{10} \sum_{a=1}^{10} \frac{1}{A + \langle w_i, x_a \rangle}. \tag{98}$$

For $m = 1$ (Fig. 8(a)), we use $A = 1.05$; for $m = 2$ (Fig. 8(b)), we use $A = 2.05$. In Fig. 8(a), we plot the error of compressing $d$ 1d random objects into $\lceil 120 \log d \rceil$; in Fig. 8(b), we plot the error of compressing $d$ 2d objects into $\lceil 20 \log^2 d \rceil$. All unspecified setting of this numerical experiment is identical with that of Fig. 2. Despite pruning bigger fraction of the objects when $d$ increases, we see that the error is not increasing with $d$, but rather overall vanishing with $d$. Thus, this shows that it is possible to compress $d$ to $O(\log^m d)$ objects losslessly. However, the numerical error shows visible oscillation, possibly due to the volatile moment-matching order $k_{\mathrm{opt}}$ and finite-size ($d$) effect.

## H    USE OF LARGE LANGUAGE MODELS

ChatGPT is used in revising the language of the main text. All ideas, theorems and derivations are formulated by the authors.

ChatGPT and OpenAI Codex are used in partially completing the code for numerical experiments. However, all the main algorithms are designed and inspected by the authors.

