# OpenReview forum: "A universal compression theory for lottery ticket hypothesis and neural scaling laws"
_ICLR.cc/2026/Conference — ICLR 2026 Poster_

### Official Review · Reviewer_LiWj · 2025-10-27

**Soundness:** 2
**Presentation:** 2
**Contribution:** 2
**Rating:** 4
**Confidence:** 4

**Summary:**

The paper studies how neural networks and datasets can be compressed by exploiting permutation symmetries. The authors show that symmetric functions can be represented using fewer variables, which implies that both the model and the data can be reduced to polylogarithmic size without significantly changing the loss. This leads to what the authors call a dynamical lottery ticket hypothesis and stronger scaling laws.

**Strengths:**

The paper presents an interesting idea: using permutation symmetry to achieve strong compression of both networks and datasets.
The theoretical argument (that symmetric functions can be represented with fewer variables), is promising. The results aim to connect model compression, scaling laws, and the lottery ticket hypothesis in a unified framework.

**Weaknesses:**

The paper proposes a theoretical link between symmetry, compression, and scaling laws. However, the lack of clear algorithmic formulation and the absence of fair experimental baselines limit its current practical relevance.

The main limitation of the paper is the lack of rigor and clarity. The compression process is described only at a high level. It is not clear how one would actually construct the compressed network or dataset in practice. The paper does not include pseudocode or complexity estimates, making it hard to evaluate the tractability of the proposed methods.

The experimental comparison is incomplete. The proposed compressed network is compared with both the original network and a random sparse network. However, it is already known that random sparse networks perform poorly, while sparse networks obtained with *Iterative Magnitude Pruning* (IMP, Frankle & Carbin 2019) can match the performance of dense ones. A fair comparison should therefore include IMP or other modern sparse training methods.

The compression-error trade-off is not clearly quantified. The claim that a network with (d) parameters can be reduced to polylogarithmic size should be expressed as a function of the error, and possibly compared to existing theoretical bounds.
Finally, some parts of the theoretical presentation are unclear. The meaning of the function $f$ in Theorem 5 is not explained, and the notation $|f' - f| = \omega(d)$ is confusing, since $ \omega(d)$ can mean any function that grows faster than $d$, but such a bound would be vacuous.

**Questions:**

- Could you provide a concrete description of the compression algorithm? How are the compressed parameters and datasets obtained from the original ones?
- How does your method compare, both in compression ratio and performance, with Iterative Magnitude Pruning or other sparse training techniques?
- Can you explicitly state the trade-off between compression and approximation error, and how it compares with previous results (e.g. to those for the Strong LTH such as Pensia et al., 2020)?
- What exactly does the function $f$ represent in Theorem 5? You didn't define it. Can you clarify the notation?
- Can you argue that the bound $|f' - f| = \omega(d)$ is not vacuous?

---

> ### Author Response · Authors · 2025-11-26
> **Reply #1 to Reviewer LiWj**
>
> We thank the reviewer for devoting time to inspect our paper. We strongly suggest you to read again and read through our paper to ensure that your assessment is consistent with our content, as the majority of points you think unclear/insufficient are in fact clearly addressed in our first draft. In the below, we provide point-to-point explanation to your concerns. Please let us know whether we properly addressed your concerns, and whether there are further improvement suggestions.
>
> **Weaknesses:**
>
> **W0. The paper proposes a theoretical link between symmetry, compression, and scaling laws. However, the lack of clear algorithmic formulation and the absence of fair experimental baselines limit its current practical relevance.**
> **A0.** Thanks for raising this point. This reason of rejecting our paper is unfair and not reasonable. For the first point, as we explain below, we do have a clear algorithmic formulation. The second point can be applied to any theory work and is not a reason for rejection at all. In fact, our contribution is only theoretical, and we never claim to make any algorithmic contribution. Our positioning of the significance of our paper is that it is the first paper to theoretically show that compression map exists which can improve scaling laws, for general deep learning tasks.
>
> **W1. The main limitation of the paper is the lack of rigor and clarity. The compression process is described only at a high level. It is not clear how one would actually construct the compressed network or dataset in practice. The paper does not include pseudocode or complexity estimates, making it hard to evaluate the tractability of the proposed methods. **
>
> **A1.** Regarding “lack of rigor”, this is not true and goes against all other reviewer evaluations. For example, Reviewer YxjE states “the authors provide both rigorous derivations and intuitive explanations for their theoretical results.” Reviewer 1dtD states “The paper delivers a rigorous theoretical result.” In fact, all our results are formulated with full standard of mathematical rigor and are based on minor well-bahavedness assumptions (see Section 2). We would be more than happy to revise and improve the manuscript if you point out concretely where the rigor and clarity problems are.
>
> Regarding clarity, we have in fact provided pseudocode and complexity estimates in the paper. The main compression algorithm is presented as pseudocode in Algorithm 1, which breaks into two parts: clustering and moment-matching. Their full detail is given in Appendix D. The moment matching algorithm is described clearly in Algorithm 2. The complexity analysis is in Section 4.2 (around line 240) and Appendix D. In the updated draft, we also added a runtime benchmark (runtime vs d) in Appendix D. This algorithm described in Appendix D is what we used to produce all numerical demonstration in Figs. 2-5.
>
>
> **Q2. The experimental comparison is incomplete. The proposed compressed network is compared with both the original network and a random sparse network. However, it is already known that random sparse networks perform poorly, while sparse networks obtained with *Iterative Magnitude Pruning* (IMP, Frankle & Carbin 2019) can match the performance of dense ones. A fair comparison should therefore include IMP or other modern sparse training methods.**
>
> **A2.** We point out that our contribution is only theoretical. All experiments are presented to validate the theoretical contributions, not intending to beat any existing neural network pruning algorithm. Moreover, a comparison between our compression and IMP is unfair and could even be misleading for they have the following differences:
>
> 1. Our compression method is applicable to a lot of machine learning modules, including MLP, transformer, and also to data pruning. However, IMP is specific to pruning neural networks.
>
> 2. IMP decides which neurons to prune by observing a few steps of training, whereas our approach is “one compression for all” which does not use information of any specific training.
>
> 3. IMP has only been empirically shown to be able to prune, colloquially, “most” of the neurons. But our paper is the first to show quantitatively what fraction of the network that can be pruned.

---

> ### Author Response · Authors · 2025-11-26
> **Reply #2 to Reviewer LiWj**
>
> **Q3. The compression-error trade-off is not clearly quantified. The claim that a network with (d) parameters can be reduced to polylogarithmic size should be expressed as a function of the error, and possibly compared to existing theoretical bounds. Finally, some parts of the theoretical presentation are unclear. The meaning of the function $f$ in Theorem 5 is not explained, and the notation $|f' - f| = \omega(d)$ is confusing, since $\omega(d)$ can mean any function that grows faster than $d$, but such a bound would be vacuous.**
>
> **A3.** This criticism is ungrounded. This tradeoff is however very clearly quantified in our main error bound theorems (Theorem 4 for finite k, and Theorem 7 for optimal k). Theorems 4 and 7 has the same structure, and we explain Theorem 4 here as an example. Eq. (9) states that if we compress d objects into d’, the error is guaranteed to be smaller than the right-hand side.
>
> Eq. (10) states that if we require the error to be smaller than $\omega(d)$, then there exists a compression map (which is our moment-matching compression, Algorithm 1) to compress d to d’, where d’ is a function of $d$ and $\omega(d)$. Indeed, Eq. (10) quantitatively suggests that the less error we allow (e.g., $\omega(d)= d^{-100}$ vs $\omega(d) = d^{-1}$), the bigger $d'$ has to be. To help better understand these bounds, we added example analysis for the case of $\omega=d^{-\alpha}$. See Eq. (11) and the newly added text below Eq. (12).
>
>
> The meaning of $f$ is clear from the context. Note in Theorem 5: “Suppose… the model prediction $f(\theta)$ is symmetric.” We have one unique definition for the notation $f(\theta)$ throughout this paper, that is, a symmetric function of $d$ variables having a finite radius of convergence (see Section 2).
>
>
> As you pointed out, $\omega(d)$ indeed causes confusion. We replaced this notation by $\varepsilon(d)$ in the new version.
>
>
> Theorems 4, 5 and 7 hold for any function $\varepsilon(d)$. But practically, it means “allowed error”, so it is useful to interpret it as satisfying $\lim_{d\to\infty} \varepsilon(d) = 0$. This is the reason why we do not specify any constraint on the function $\varepsilon(d)$ in our theorem statements. This resembles the $\varepsilon$-$\delta$ notation in calculus: those statements hold for any positive $\varepsilon$, but are informative only when $\varepsilon$ approaches zero.
>
>
> **Questions:**
>
>
> **Q4. Could you provide a concrete description of the compression algorithm? How are the compressed parameters and datasets obtained from the original ones?**
>
>
> **A4.** See our explanation in [A1], and we kindly suggest you to read Appendix D. Please indicate which parts need to made clearer, if any.
>
> **Q5. How does your method compare, both in compression ratio and performance, with Iterative Magnitude Pruning or other sparse training techniques?**
>
> **A5.** Again, we believe that comparing performance with existing algorithms is unnecessary for a theory paper, and we do not intend to beat any specific existing algorithm. Our main contribution is the theory establishing that (with some qualifiers): “Permutation-symmetric functions can be compressed to Polylog,” establishing this claim does not require any algorithm comparison. Also, see our explanation in [A2].
>
> **Q6a. Can you explicitly state the trade-off between compression and approximation error...**
>
> **A6.** Please see our explanation in [A3].
>
> **Q6b: ...and how it compares with previous results (e.g. to those for the Strong LTH such as Pensia et al., 2020)?**
>
> **A6b.** Our theory is different and not comparable to theirs at all. In fact, this explanation has been made in our original manuscript in line 139-145—we added Pensia et al. to reference and expanded this discussion to make the difference clearer.
> The theorem proved in Pensia et al., 2020 (and other Strong-LTH-related results) is very different from ours (and, arguably, the original LTH). Our theorem and the original LTH makes a clear hypothesis about the learning dynamics being the same before and after compression (see the quotation of the original LTH below). In contrast, the strong LTH has no guarantee about the learning dynamics (see quotation below). Therefore, we are proving two different kinds of results that are not comparable.
> Because our theory is the only provable version of LTH that explicitly shows the invariance of the learning dynamics after compression, our result is called the “dynamical LTH.”
>
> The previous works (Malach et al., 2020; Pensia et al., 2020; da Cunha et al., 2022) prove the “strong LTH”. The strong LTH is about representational richness of overparametrized networks, which does not involve training at all. For your reference, here is a comparison between the statement of LTH and the strong LTH.

---

> > ### Comment · Reviewer_LiWj · 2025-11-27
> >
> > I thank the authors for their detailed rebuttal. I appreciate the clarifications regarding the algorithmic details in Appendix, the complexity estimates , and the quantitative trade-off.
> >
> > While I acknowledge the theoretical nature of the work, I disagree with the assertion that the lack of fair experimental baselines is "not a reason for rejection at all" for any theory paper. Assessing the tractability and practical implications of a theoretical concept through rigorous, comparative experiments is a standard and valuable component of ICLR submissions.
> >
> > However, the authors have sufficiently addressed my other specific concerns.
> > Based on the clarifications provided, I am updating my score.

---

### Official Review · Reviewer_ui4S · 2025-10-30

**Soundness:** 3
**Presentation:** 4
**Contribution:** 3
**Rating:** 8
**Confidence:** 2

**Summary:**

The paper introduces the universal compression theorem as a step towards the dynamical lottery ticket hypothesis (LTH), which claims that in a dense network there exists a subnetwork, which when trained in isolation exhibits the same training dynamics as the original one. The theorem states (informally) that a permutation-invariant function of $d$ variables each of dimensionality $m$ can be asymptotically compressed to a function of $O(\text{polylog } d)$ variables. The authors argue that, because many model / dataset objects are symmetric in parameters / datapoints, these results imply polylog-rate network and dataset compression under the assumptions of the theorem.  Another implication of polylog compression is the scaling law $L \approx L_0 + C d ^{-\alpha}$ changing from power law form to stretched-exponential form  $L \approx L_0 + \exp (- \alpha’  \sqrt[m]{d})$, both for model and dataset size.

**Strengths:**

1. The paper provides theoretical grantees on asymptotic polylogarithmic compression for symmetrical functions. The authors provide Algorithm 1 for compression of symmetric functions using moment-matching and validate it numerically.
2. An important feature is the universality of the result: the implications of the theorem include both neural networks and datasets.
3. A major practical consequence of the work is the potential speed up guarantees on the power-law scaling laws, which are known to "be slow", i.e. have small power exponentials.
4. Although the main result is theoretical, the authors back each claim with numerical experiments: they show on a synthetic function that compression error drops with in agreement with the theoretical bound (Fig. 2); that training dynamics on a compressed dataset follows training on the full dataset (Fig. 3); training performances of full and compressed models are identical to support dynamical LTH (Fig. 4); and compressing a network or dataset leads to a larger scaling law exponent (Fig. 5). These comprehensive validations neatly complement the theoretical backbone of the paper.

**Weaknesses:**

1. Further empirical evaluation would strengthen this work, as the authors note.
2. The proposed moment-matching algorithm scales poorly with moment order $k$ and dimension $m$ (via $\binom{m+k}{k}$), which limits immediate practical effects despite the asymptotic guarantees.
3. The theoretical claim of polylogarithmic compression yielding a stretched-exponential scaling $\text{exp} (- \sqrt[m]{d})$ is not supported with evidence. The numerical experiments in Section 6 demonstrate how the scaling laws can be improved only for quadratic compression.

**Questions:**

1. Can you show an example with the scaling laws of a form $L \approx L_0 + c \text{exp} (- \alpha’ \sqrt[m]{d})$ to illustrate the stretched-exponential regime?
2. In numerical experiments in Section 6 the exponent should have improved by a factor of 2: $C d^{-\alpha} = C (\frac{d’}{16})^{-2 \alpha} =C’ (d’)^{-2\alpha} $. The reported values are close but lower, 1.271 vs $2\alpha = 1.366$ and 0.608 vs $2 \alpha=0.616$. Why does this difference appear? And why is it larger for dataset compression?
3. Many elements of modern neural networks do not fall under the smoothness assumptions, like ReLU, top-k selections, sparse \ quantized representations. How do you imagine expanding your work around those limitations and how would compression rates be affected?

---

> ### Author Response · Authors · 2025-11-26
> **Reply #1 to Reviewer ui4S**
>
> We appreciate the reviewer’s effort and constructive comments which helped us improve the text. In this reply, we address the reviewer’s concerns point by point.
>
> **Weaknesses:**
>
> **W1. Further empirical evaluation would strengthen this work, as the authors note.**
> **A1.** In the revised paper, we added the following numerical experiments:
> 1. Compressing an attention module and showing LTH-like consistency (Appendix F) (related to questions from Reviewers 1dtD and YxjE)
> 2. Compression algorithm runtime vs d (Appendix D) (related to questions from Reviewer 1dtD)
> 3. Demonstration of the ability to compress d to log(d) when $m=1$ or $2$ (Appendix G) (related to questions from Reviewers YxjE and ui4S)
>
>
> **W2. The proposed moment-matching algorithm scales poorly with moment order $k$ and dimension $m$ (via $\binom{m+k}{k}$), which limits immediate practical effects despite the asymptotic guarantees.**
> **A2.** Thanks for this criticism, we agree. However, we would like to emphasize that our primary contribution is theoretical, and the method we suggested primarily serves as part of the constructive proof and a proof of principle. Our theory motivates the search for more efficient ways to compress models and data. We believe these are important future works.
>
> **W3. The theoretical claim of polylogarithmic compression yielding a stretched-exponential scaling $\text{exp} (- \sqrt[m]{d})$ is not supported with evidence. The numerical experiments in Section 6 demonstrate how the scaling laws can be improved only for quadratic compression.**
>
> **A3.** In fact, our FIg. 2 is intended to show that the compression residual error vanishes as expected. In the revised paper, we added a numerical experiment that compresses $d$ to $\operatorname{polylog}(d)$, which supports the ability to improve power-law scaling laws to stretched-exponential. However, we admit that compressing to polylog(d) requires tremendous resource for our current algorithm, and hence we are not able to quantitatively compare the residual error to theoretical predictions as we did for finite k in Fig. 2. The errors are shown to be overall vanishing, but also shows visible oscillation, possibly due to the volatile moment-matching order $k_{\mathrm{opt}}$ and finite-size ($d$) effect.
>
>
>
> **Questions:**
>
> **Q4. Can you show an example with the scaling laws of a form $L \approx L_0 + c \text{exp} (- \alpha’ \sqrt[m]{d})$ to illustrate the stretched-exponential regime?**
> **A4.** See our reply in **[A3]**.
>
> **Q5. In numerical experiments in Section 6 the exponent should have improved by a factor of 2: $C d^{-\alpha} = C (\frac{d’}{16})^{-2 \alpha} =C’ (d’)^{-2\alpha} $. The reported values are close but lower, 1.271 vs $2\alpha = 1.366$ and 0.608 vs $2 \alpha=0.616$. Why does this difference appear? And why is it larger for dataset compression?**
>
> **A5.** Thanks for this question. First of all, this is a rather small deviation, and as in any empirical science, the theoretical values will have some deviation from the empirical results due to, for example, systematic errors in the experiments. The main reason of this deviation is a finite-size ($d$) effect. Concretely, compression induces an error $d^{-\beta}$ where $\beta>\alpha$. But at finite $d$ it still affects the observed scaling law. Another possible source of error is the numerical precision of the FP32 format. Testing the actual reasons of these deviations is an interesting future problem.
>
>
>
> **Q6. Many elements of modern neural networks do not fall under the smoothness assumptions, like ReLU, top-k selections, sparse \ quantized representations. How do you imagine expanding your work around those limitations and how would compression rates be affected?**
>
> **A6.** Thanks for this interesting question. Extending the theory to functions with a limited smoothness is an important future step. There are conventional wisdoms of how smoothness is related to how compressible or approximatable a function is (sometimes known as the blessing of smoothness). Some works imply that the best compression rate is $d^{-k}$ if the network uses ReLU^k as the activation (e.g., doi.org/10.1007/s00211-023-01384-6). However, a unified theory linking generic non-smoothness to compression is an open problem beyond our scope.

---

> > ### Comment · Reviewer_ui4S · 2025-11-26
> >
> > I thank the authors for providing detailed responses and conducting additional experiments. I am satisfied with the answers and will keep my score at 8.

---

### Official Review · Reviewer_1dtD · 2025-10-31

**Soundness:** 3
**Presentation:** 4
**Contribution:** 4
**Rating:** 6
**Confidence:** 3

**Summary:**

The paper proves a universal compression theorem, showing that almost any symmetric function of $d$ elements can be compressed to a function with $O({\rm polylog}$ (d)) elements losslessly. The theory leads to two key applications. First is the dynamical lottery ticket hypothesis, proving that large networks can be compressed to polylogarithmic width while preserving their training dynamics. Second is dataset compression, demonstrating that neural scaling laws can be theoretically improved from power-law to stretched-exponential decay.

**Strengths:**

- The paper delivers a rigorous theoretical result that proves the dynamical lottery ticket hypothesis by showing that large networks can be compressed while preserving their original training dynamics.
- Provides a generalized compression theory with broad applicability across diverse domains (e.g., dataset and model compression), demonstrating strong theoretical versatility and significant potential for cross-domain impact.
- Establishes clear practical advantages, such as improved scaling laws and model compression, that are well grounded in the proposed theoretical framework.

**Weaknesses:**

- The paper lacks a thorough discussion on the applicability of the proposed theory to complex neural architectures such as Transformer blocks, which integrate linear projections, attention mechanisms, and normalization layers.
- There seems to be a missing reference link to the Appendix at line 190 on page 4 (“Appendix ??”).

**Questions:**

- The model assumes neuron permutation symmetry. Does the assumption is applicable to complex modules in neural networks, such as Transformer block?
- In experiments such as Figure 3 or 4, how much real computation time does the proposed compression take?

---

> ### Author Response · Authors · 2025-11-26
> **Reply #1 to Reviewer 1dtD**
>
> We appreciate the reviewer’s effort and constructive comments which helped us improve the text. In this reply, we address the reviewer’s concerns point by point.
>
>
> **Weaknesses:**
>
> **W1. The paper lacks a thorough discussion on the applicability of the proposed theory to complex neural architectures such as Transformer blocks, which integrate linear projections, attention mechanisms, and normalization layers.**
> **A1.** Our compression theory can be applied to attention layers in two different ways. We briefly mentioned these points in Section 2, around line 101; in the revised draft, we added the new Appendix F to expand the discussion on transformers, including a minimal numerical showcase of the effectiveness of compressing transformers.
>
> The first is a rather trivial application to compressing query and key matrices, and the second is the more interesting and complicated compression of attention heads.
> The first is a trivial application to the key and query weight matrices $W_Q$ and $W_K$ – which is a good sanity check for our theory. The output of the attention logit depends on the product of the two matrices: $a=a(W_Q W_K)$, notice that one can write this product as the following sum of outer product:
> $$\sum_i^d  w_Q^i (w_K^i)^T$$
> where $w_Q^i$ is the $i$-th row of $W_Q$ and $w_K^i$ is the $i$-th column of $W_K$. The width $d$ corresponds to the right dimension of $W_Q$. This means that there is a permutation symmetry: one can permute the orders of $i$ because it is a dummy index. This implies that we can compress these rows and columns together – but this is already obvious from linear algebra, if the left dimension of $W_Q$ is $m$, then $W_Q W_K$ is at most rank $m$, and thus, it is not useful to for $d$ to be larger than $m$. One can thus achieve an $O(polylog(d))$ compression (in fact, it is not just polylog, but a constant compression). Therefore, this is a good sanity check of the correctness of the theory.
> This argument is much more interesting when one tries to make the key-query matrices nonlinear, which could be an interesting future direction. For example, one can define a nonlinear function $s$ such that QK product is replaced by
> $$\sum_i^d  w_Q^i s(w_K^i, X)^T$$
> where $X$ is the data. Now, our theory immediately implies that one can in principle achieve a PolyLog compression for this attention layer.
> Now, let us consider the second type, the compression of attention heads. Again, it suffices to identify where the permutation symmetry is. Consider a layer with $d$ attention heads and $A_i = B(w_i,X)$ denote the $i$-th head, and $w_i$ is its trainable parameter. Following the attention heads, one often performs the following computation:
> $$Output = U \, concat(A_1,...,A_d)$$,
> where $U \in \mathbb{R}^{z \times dh}$ is an output matrix of the entire attention layer, and $h$ is dimension of each attention head output. This output can be written as
> $$Output = \sum_i^d U_i B(w_i,X) $$
> where $U_i  \in \mathbb{R}^{z \times h}$ is the block of $U$ that takes the output of $A_i$ as the input. This summation structure makes clear that there is a permutation symmetry between the parameters $\theta_i = (U_i, w_i)$, and so $\theta_i$ can be compressed to $Polylog(d)$ according to our theory.
>
>
>
>
> **W2. There seems to be a missing reference link to the Appendix at line 190 on page 4 (“Appendix ??”).**
>
>
> **A2.** Thanks for pointing this out. We fixed it by referring to Appendix D where we describe the entire moment matching algorithm in detail.
>
> **Questions**
>
> **Q3. The model assumes neuron permutation symmetry. Does the assumption is applicable to complex modules in neural networks, such as Transformer block?**
>
> **A3.** Permutation symmetry can arise from the parameters in attention modules as well. As we briefly mentioned around line 101: “attention logits in self-attention, and attention outputs between attention heads”. We expanded this part to describe the permutation symmetry in attention in Section 2. We demonstrate compressing transformers in the new Fig. 7. See our answer to W1 above.
>
>
>
>
> **Q4. In experiments such as Figure 3 or 4, how much real computation time does the proposed compression take?**
>
> **A4.** We added numerical result of the runtime vs $d$ in Appendix D “Details on numerical experiments” with discussions. Theoretical analysis of the runtime scaling is presented in this Appendix as well in our first draft.

---

> > ### Comment · Reviewer_1dtD · 2025-11-27
> >
> > I appreciate the authors’ feedback and their detailed responses to my questions. I will keep my score.

---

### Official Review · Reviewer_YxjE · 2025-11-01

**Soundness:** 3
**Presentation:** 3
**Contribution:** 2
**Rating:** 4
**Confidence:** 3

**Summary:**

This work addresses dataset and neural network compression from a moment-matching perspective. Under certain assumptions, this approach establishes novel compression rates and power laws for these tasks. It also enables the boosting of neural power laws, which describe performance versus dataset size dynamics. A number of low-dimensional experiments are conducted to support the claims.

**Strengths:**

The work is mathematically sound and easy to follow. The text is clear, supported by a decent and concise background overview. The authors provide both rigorous derivations and intuitive explanations for their theoretical results, and the experiments support their claims across a number of settings.

**Weaknesses:**

My main criticism revolves around the **curse of dimensionality**, which the authors underaddress several times throughout the paper.

1. Both (9) and (10) have dimensionality-dependent exponents, which explode when $m \to \infty$ given that other constants are fixed. This is later combated by selecting $k > (1 = \sigma^{-1}) m - 1$, which, in turn, explodes $\binom{m+k}{k}$. Through some trickery in Theorem 7 (unfortunately, due to time constraints, I was not able to fully verify the math), the authors miraculously balance these issues by attaining a poly-log compression rate.

    That said, one might expect that substituting $d'$ from (45) into (44) should yield errors which are (asymptotically) under some fixed $\omega$. However, when done numerically for $m=10$, $\rho=0.1$, $\omega=0.1$, and any multiplicative constant in (45), I always get an exploding upper bound on the compression error. Reasonable variations of $\rho$ and $\omega$ do not alleviate the issue, which only worsens as $m$ grows.

2. Since $k$ in Theorem 7 grows with increasing $d$, $f$ is required to be increasingly smooth. While most contemporary NNs are $\infty$-smooth almost everywhere, their numerical smoothness degrades with increasing dimensionality or a decreasing learning rate [1]. In practice, this will take a toll on the derived bounds in terms of asymptotic constants or other parameters (e.g., $\rho$ in (44)). This problem remains unaddressed in the main text.

3. The experimental setups are toy, with the dimensionality being $4-12$ orders of magnitude lower than in real-world tasks. In my opinion, this might lead to the following problems:
    - While showing decent performance in low-dimensional regimes, the proposed compression method might entail overfitting in high-dimensional setups. Stochastic gradient descent (SGD) is known to apply implicit regularization during training [2], thus selecting less overfitting solutions. Your method, however, might "overcompress" a NN/dataset: among all solutions, a non-generalizable one is selected (train error or even dynamics are the same, but test error is not).
    - It is known that some problems in ML have exponential (in dimensionality) sample complexity (e.g., density estimation). Your result, however, suggests that these problems are also log-exponential in dimensionality (Theorem 7 applied to dataset compression) given the train error is preserved. The only logical conclusion I can arrive at is that such compression almost always entails overfitting when considering complex problems.

4. While the authors briefly mention the manifold hypothesis in Section 7, it is not clear how one can use it to improve the method. Moment matching is agnostic to manifolds: i.e., it generally cannot capture such intricate structures. Therefore, another manifold learning strategy must be employed beforehand to decrease the dimensionality. Such a strategy typically requires the full dataset, as manifold learning is usually of exponential sample complexity.

[1] Cohen et al. "Gradient Descent on Neural Networks Typically Occurs at the Edge of Stability". Proc. of ICLR 2021.

[2] Smith et al. "On the Origin of Implicit Regularization in Stochastic Gradient Descent". Proc. of ICLR 2021.

**Minor issues:**

1. Broken reference in line 190: "Appendix ??"

**Questions:**

1. Can you, please, provide additional experiments (e.g., for high dataset dimensionality or low sampling sizes) proving that your method avoids overfitting?
2. I kindly ask to address my concerns in Weakness 1. In particular, I am interested in the numerical verification of the bounds provided.

---

> ### Author Response · Authors · 2025-11-26
> **Reply #1 to Reviewer YxjE**
>
> We appreciate the reviewer’s effort and constructive comments which helped us improve the text. In this reply, we further justify the existence of the curse of dimensionality, and address the reviewer’s concerns point by point. Please let us know whether we properly addressed your concerns, and whether there are further improvement suggestions.
>
>
> Overall, the curse of dimensionality appears ubiquitously in machine learning, and is present in our compression theory as well. Here, the curse of dimensionality (when m=dimension of each symmetric component becomes larger) takes the form of (a) the compressed dataset size $d’$ reachable by moment-matching is larger (b) the theoretical lower bound $d’=\log^m(d)$ is larger (c) the complexity of moment matching grows as power law of $\binom{m+k}{k}$. The main argument we want to show here is that, the curse of dimensionality is not a shortage of our particular compression algorithm, but rather a theoretical limitation on any possible universal compression. Our compression algorithm is actually optimal up to a constant.
>
>
> Our positioning of the significance of our paper is that it is the first paper to theoretically show that compression map exists which can improve scaling laws, for general deep learning tasks. As pointed out by the reviewer, it involves some subroutines that lack practicality (e.g., finding cluster with smallest diameter is in fact NP-hard), but they in turn give theoretically guaranteed error upper bounds. We do expect that future works can propose more efficient compression.
>
> **Weaknesses:**
>
> **W1a. “Both (9) and (10) have dimensionality-dependent exponents, which explode when $m \to \infty$ given that other constants are fixed. This is later combated by selecting $k > (1 = \sigma^{-1}) m - 1$, which, in turn, explodes $\binom{m+k}{k}$. Through some trickery in Theorem 7 (unfortunately, due to time constraints, I was not able to fully verify the math), the authors miraculously balance these issues by attaining a poly-log compression rate.”**
>
> **A1a.** Regarding the error bounds Eqs. (9) and (10), their dependence on $m$ is a reasonable justification of the curse of dimensionality. Especially, Eq. (10) (error ~ $\log^m d$) matches the lower bound, so it is impossible to further improve (for a **universal** compression map; for tasks with hidden structure it is possible to improve).
>
>
>  “$N_{m,k} = \binom{m+k}{k}$ explodes” mainly influences the complexity of our compression algorithm. In the clustering subroutine, finding clusters larger than $N_{m,k}$ is increasingly hard (generally, finding a cluster of points with the smallest distance is NP-hard), and the moment-matching subroutine, we do linear algebra in dimension $\sim N_{m,k}$, which brings $\operatorname{poly} N_{m,k}$ complexity. Hence, our compression algorithm can become too complex to implement. Our paper is intended to show that universal compression algorithm exists, but finding more efficient approaches is indeed an important future direction.
>
> In Theorem 7, we did not remove the exploding dependence on $m$ and $k$. Here, we explain the logical connection between Theorems 4 and 7. In this paper, $d$ (the number of symmetric objects) is always the biggest scale: $1\ll m,k \ll d$. In a finite-$k$ algorithm, nothing depends on $d$ except the number of clusters we need to handle. This regime gives us the power law in Theorem 4 and the seemingly big $\binom{m+k}{k}$. For all finite $k$, larger $k$ yields smaller error, and the optimal value is $k_{\mathrm{opt}} \sim d'^{1/m}$, which is scaling up with $d$. So although in Theorem 7 $\binom{m+k}{k}$ does not appear, $k_{\mathrm{opt}}$ is by assumption an even larger value.
>
> **W1b. “That said, one might expect that substituting $d'$ from (45) into (44) should yield errors which are (asymptotically) under some fixed $\omega$. However, when done numerically for $m=10$, $\rho=0.1$, $\omega=0.1$, and any multiplicative constant in (45), I always get an exploding upper bound on the compression error. Reasonable variations of $\rho$ and $\omega$ do not alleviate the issue, which only worsens as $m$ grows.”**
>
> **A1b.** To alleviate confusion about how to interpret the $\varepsilon(d)$ (used to be $\omega(d)$; we changed notation) bounds, we added case studies on $\varepsilon(d) = d^{-\alpha}$ in the main text (see Eq. (11) and the discussion below Eq. (12)). For your specific question, let’s say $d' = A\log d$. Putting this into Eq. (45), we get $\mathcal{E} = O( (\log d)^{m-1} d^{1 - A(m!\rho)^{1/m}/e} )$. $(\log d)^{m-1}$ can be ignored; by choosing sufficiently large $A$ we can make this error to be smaller than any power law. So Eqs. (44) and (45) are consistent.

---

> ### Author Response · Authors · 2025-11-26
> **Reply #2 to Reviewer YxjE**
>
> **W2. “Since $k$ in Theorem 7 grows with increasing $d$, $f$ is required to be increasingly smooth. While most contemporary NNs are $\infty$-smooth almost everywhere, their numerical smoothness degrades with increasing dimensionality or a decreasing learning rate [1]. In practice, this will take a toll on the derived bounds in terms of asymptotic constants or other parameters (e.g., $\rho$ in (44)). This problem remains unaddressed in the main text.”**
>
> **A2.** While we appreciate your thoughtful comment, we do not think it is the right logic to evaluate our work. We would like to first emphasize that our paper is a mathematical theory, and this contribution is independent of whether it is useful or not at practical “numerical smoothness”. As an analogy, it makes no scientific sense to say things like “the Riemann Hypothesis suffers from finite precision of computers” because this criticism is unrelated to the theoretical value of proving the Riemann hypothesis.
> Throughout the paper, our only “well-behavedness” assumption for the function $f$ is that it has a finite convergence radius, which means that the Taylor expansion scales slower than some power law $\rho^{-k}$. There are conventional wisdoms of how smoothness is related to how compressible or approximatable a function is (sometimes known as the blessing of smoothness). Indeed, according to [1], this assumption is likely to break down when $m$ scales up. We are mainly focusing on finite $m$, but as the audience might be interested in extending to large $m$, in the revised paper, we addressed this possible non-smoothness issue in our Discussion and Outlook.
>
>
> **W3a. “The experimental setups are toy, with the dimensionality being $4-12$ orders of magnitude lower than in real-world tasks. In my opinion, this might lead to the following problems:”**
>
> **A3a.** Real-life tasks indeed have much larger dimension than our numerical examples, but they are not as big as 8-12 orders of magnitude bigger, because we are always implicitly exploiting hidden structures of seemingly high-dimensional problems (e.g., local correlation in images, low entanglement in quantum states). Despite we proved that the compression rate to $\log^m d$ is optimal, we do expect that future works can develop much more efficient compression algorithms which can make compression in $m\sim 10^3$ actually practical. We deem this is an important future direction.
>
> **W3b. “While showing decent performance in low-dimensional regimes, the proposed compression method might entail overfitting in high-dimensional setups. Stochastic gradient descent (SGD) is known to apply implicit regularization during training [2], thus selecting less overfitting solutions. Your method, however, might "overcompress" a NN/dataset: among all solutions, a non-generalizable one is selected (train error or even dynamics are the same, but test error is not).”**
>
> **A3b.** To the best of our knowledge, there is no logical connection between compression and overfitting. Both the train error and the test error are symmetric functions of the dataset/model parameters. So our error bounds (Theorems 4 and 7) apply to both of them simultaneously. For larger $m$, there is still no reason why they can bypass our error bounds (although we cannot demonstrate it for reasons explained in **[A6]**).
>
> Thus, overfitting can happen, but it happens only when the model before compression is overfitting—it is not a problem of our compression but a problem of the original model. The statement we proved is: a $d$-variable function outputs essentially identical to the compressed $d’$-variable function. Sometimes $d$ can be too large so it overfits (memorizing instead of generalizing)—in this case the compressed $d'$ also overfits. Hence, large $d$ can overfit compared to naive small $d'$, which is common in ML. Compression is particularly useful when there is no overfitting and the loss scales to zero as $d\to\infty$.

---

> ### Author Response · Authors · 2025-11-26
> **Reply #3 to Reviewer YxjE**
>
> **W3c. It is known that some problems in ML have exponential (in dimensionality) sample complexity (e.g., density estimation). Your result, however, suggests that these problems are also log-exponential in dimensionality (Theorem 7 applied to dataset compression) given the train error is preserved. The only logical conclusion I can arrive at is that such compression almost always entails overfitting when considering complex problems.**
>
> **A3c.** This is in fact an interesting point and common misunderstanding. The existence of a $d\to \log^m d$ compression *per se* does not imply that we can reduce the sample complexity. Let’s say density estimation which is proven to have exponential sample complexity ($d=\Omega(e^{cm})$). We can simplify the procedure of
>
> - Sample $d$ points -> construct estimator $\hat{f}(x_1, \dots, x_d)$
>
>
> to
>
> - Sample $d$ points -> compress to $d'$ weighted points $(c_j, w_j)$ (j=1,2,..., d') -> construct estimator $\hat{f}(\{(c_j, x_j)\}_{j=1}^{d'})$
>
>
> To be specific, we cannot get the $d'$ weighted points out of nowhere.  The estimator construction can be simpler, but this does not reduce the sample complexity.
>
> However, our theory could imply novel strategies on sampling. We mentioned this idea in Discussion and Outlook, and deem that it is an important direction we should pursue in the future. It does not work for density estimation, but works for many supervised learning tasks. For example, the task is to learn a function $f(x)$ from samples $(x, f(x))$, where we can decide what $x$ to query. Then a clever strategy is to make the dataset maximally “hard to compress.” This idea could be related to importance sampling or some physical experiment strategies (when I think the curve is smooth, I sample less; when there seems to be a cusp, I sample more).
>
>
>
>
> **W4. While the authors briefly mention the manifold hypothesis in Section 7, it is not clear how one can use it to improve the method. Moment matching is agnostic to manifolds: i.e., it generally cannot capture such intricate structures. Therefore, another manifold learning strategy must be employed beforehand to decrease the dimensionality. Such a strategy typically requires the full dataset, as manifold learning is usually of exponential sample complexity.**
>
> **A4.** This is a very good question. Even without performing manifold learning, the fact that the data lies on a low dimensional manifold still helps. There are two ways where a low dimensionality helps:
>
> ​	(1) The error rate of the algorithm: this only depends on the intrinsic dimension of the data, and automatically improves if the data lies on a low dimensional manifold. This is because the error bounds depends on dimension only through the distance between data points, and points are likely to be closer to each other if they lie on a low-dimensional manifold. Hence, Theorems 4 and 7 etc. all automatically have a better rate, even if we do not know a description of the manifold.
>
> ​	(2) The computation and memory complexity of the algorithm: this part only improves if we explicitly reduce the dimension, and we agree that some form of manifold learning is required. Concretely, knowing a manifold helps in the following way. Suppose each $w_i\in\mathbb{R}^m$, and we know that it can be parametrized by fewer scalars: $w_i = w(x_i), x_i\in \mathbb{R}^{m'}$, where $m'<m$. Then we can view the original symmetric function as a symmetric function of $x$’s. It follows that for any fixed $k$, the complexity of moment-matching in $m’$ dimensions is smaller than in $m$ dimensions. In the revised paper, we added description of the above mechanism in Appendix D.
>
> As the reviewer pointed out, learning a lowest-dimensional manifold is hard—in fact, at least as hard as learning any specific property of the dataset. On one hand, the fact that data lies on a low-dimensional manifold without knowing $w(\cdot)$ already helps, because the sphere packing theorem (Lemma 1 in Appendix A.4) now guarantees that the smallest cluster has radius $O(d^{-1/m'})$ instead of $O(d^{-1/m})$. So although the moment matching step is still hard to implement, the error bounds (Theorems 4 and 7) are improved as replacing all $m$ by $m'$. Knowing the function $w(\cdot)$ further reduces the complexity of the moment matching step of our algorithm. On the other hand, there is a tradeoff: we can make fewer effort to learn some partial structure of the dataset and use it to reduce dimension, and then feed the smaller-dimensional problem to a compressor and a machine learning model. For example, if we want to learn properties of a stochastic process, its degree of freedom is generally exp(length)—extremely high. But if we priorly know, or learn from a few examples, that the process is Markovian, then we can parametrize it by a set of transfer matrices, which has only $\propto$ length degrees of freedom. Language data also appear to moderate effective dimensions (see Discussion and Outlook).

---

> ### Author Response · Authors · 2025-11-26
> **Reply #4 to Reviewer YxjE**
>
> **Minor issues:**
>
>
> **Q5. Broken reference in line 190: "Appendix ??”**
>
>
> **A5.** Thanks for pointing this out. We fixed it by referring to Appendix D where we describe the entire moment matching algorithm in detail.
>
>
> **Questions:**
>
>
> **Q6. Can you, please, provide additional experiments (e.g., for high dataset dimensionality or low sampling sizes) proving that your method avoids overfitting?**
>
> **A6.** As explained in **[A3]**, overfitting can happen but is irrelevant to our algorithm at all. This is because the model before and after compression is exactly the same. The fact that there is no such difference before and after compression is clear from Figure 4. For your information, the reason why we did not show the train loss in Fig. 3 is because in that experiment it is basically identical to the test loss curve.
>
> Our ability to perform numerical experiments in larger dimensions is unfortunately limited by resource. For example, $m=100$. By Theorem 4, $k$ needs to be at the same order as $m$ to get a vanishing error. Let’s say $k=100$. Then $\binom{m+k}{k} \approx 10^{59}$. This means that we are guaranteed to be able to compress losslessly when $d\gg 10^{59}$, which, although sounds formidable, is still consistent as we always set $d$ to be the largest scale in our problem.
>
> Nevertheless, in the new draft, we added an example of compressing a multi-head attention module which has larger effective $m$ than any of the existing experiments, which is $m=8$. See the new Appendix F.
>
>
>
> **Q7. I kindly ask to address my concerns in Weakness 1. In particular, I am interested in the numerical verification of the bounds provided.**
>
> **A7.** The power-law bound derived in Theorem 4 is well verified in Fig. 2. We conducted a new numerical verification for $m=1$ and $m=2$, showing that compressing to $\log^m d$ is indeed possible, which is presented in Appendix G. However, we admit that compressing to polylog(d) requires tremendous resource for our current algorithm, and hence we are not able to quantitatively compare the residual error to theoretical predictions as we did for finite k in Fig. 2. The errors are shown to be overall vanishing, but also shows visible oscillation, possibly due to the volatile moment-matching order $k_{\mathrm{opt}}$ and finite-size ($d$) effect.

---

> ### Comment · Reviewer_YxjE · 2025-11-26
> **Thank you**
>
> Dear Authors,
>
> Thank you for your detailed response. I will provide a full point-by-point reply in due course, but for now, I would like to address the most critical point of contention.
>
> 1. **Numerical non-smoothness**
>
>    I respectfully disagree with the authors. In Deep Learning, we deal with functions that can become arbitrarily ill-posed; indeed, they often do so even when acquired through SGD-based optimization. Therefore, it is not theoretically sound to assume that the radius of convergence, $\rho$ (for function $g$ in Eq. 44; *please, also note the notational conflict with $\rho$ in Eq. 4*), is constant with respect to $m$. On the contrary, [1] suggests that $\rho$ may decrease rapidly with increasing $m$. This directly impacts the asymptotics in Eq. 44, which is a theoretical, not merely a practical, problem.
>
> 2. **Overfitting**
>
>    Let me clarify my concern. As I understand it, in practice, the network is compressed to preserve its performance on the **training data**, as using evaluation data would lead to information leakage. The text appears to provide no theoretical guarantees on the evaluation loss when the symmetric function is defined solely through the training set. I suspect such guarantees are impossible to provide because the primary driver of good generalization in overparameterized regimes is the implicit biases of NNs and SGD, which is not respected by the proposed compression method. I.e., one can not achieve good compression on the test set without having this set in the first place.
>
>    This logic also applies to the scaling laws. For instance, (Henighan et al., 2020) derive empirical laws connecting evaluation performance to factors like network size. While one could plug this paper's results into such laws, we can only guarantee that the training performance will be maintained, not the evaluation performance.

---

> ### Author Response · Authors · 2025-11-27
> **Reply (round 2, part 1) to Reviewer YxjE**
>
> Thank you for your prompt engagement, and for giving us the opportunity to clarify the role of smoothness and overfitting in our work.
>
> **Q1. Numerical non-smoothness**
>
> **A1.** Thanks for point out the notational conflict. We changed the function $\rho$ in the deep-set representation Eq. (4) to $h$.
>
> Below, we analyze the relevance of Cohen et al. [1] to our theory. Our conclusion is that [1] does not provide evidence that analyticity radius of an NN-represented function collapses with dimension; instead, it only documents curvature vs optimization time dynamics. Our work makes no assumptions that $\rho$ is independent of $m$. Eq. (44) precisely characterizes how the error scales with $\rho$, even if $\rho$ decreases with $m$.
>
> - What Ref. [1] actually establishes
>
>   Ref. [1] empirically shows that during *gradient descent training*, the loss Hessian typically evolves toward the **edge-of-stability** regime, where the largest eigenvalue approaches the stability threshold $2/\eta$, and afterward stays around $2/\eta$. Crucially, Ref. [1] does **not** claim that the underlying function becomes provably non-analytic, nor that its analyticity radius (or any notion of Taylor-series convergence radius) shrinks with dimension m. Even if we colloquially say that larger Hessian suggests that $f$ is likely to be less smooth, Ref. [1] only describes how smoothness depends on the learning rate—no dependence on $m$ mentioned. The observations in [1] describe a phenomenon of **training dynamics** (particularly, the learning rate in SGD controls the numerical non-smoothness), not a theorem relating the input dimension to the breakdown of analytic structure. Thus, “$\rho$ decreases rapidly with m” is not a conclusion of Ref. [1].
>
> - Dependence on $m$
>
>   It is mathematically true that for some function classes the analyticity radius can deteriorate with increasing m, unless the function class we are interested in is specially structured. This is well known in high-dimensional approximation theory and is unrelated to the phenomenon reported in Ref. [1]. Our theorems explicitly quantify how the constants in Eq. (44) depend on $\rho$; no step of the argument assumes that $\rho$ is independent of $m$. If future work identifies practically relevant classes of high-dimensional functions for which analyticity deteriorates with dimension, then we can plug in $\rho(m)$, and the bounds in Eq. (44) would degrade in the *standard, explicitly parametrized* way already visible in our formulas. This is the expected behavior of any analyticity-based compression theorem when applied to functions whose analyticity radius collapses. We will add a discussion of this point to our final revision.

---

> ### Author Response · Authors · 2025-11-27
> **Reply (round 2, part 2) to Reviewer YxjE**
>
> **Q2. Overfitting**
>
> **A2.** “The network is compressed to preserve its performance on the **training data**”— this is not ture and a severe underestimation of the strength of our theory. To the contrary, we proved that there exists a **single** compression map that simultaneously can keep the value of **virtually any symmetric function** unchanged (as long as they satisfy the smoothness conditions we required). In a different word, the bound we established is a uniform bound for all (smooth) symmetric functions. Although we always meant so, our Theorems 4, 5 and 7 might have not made it clear enough. We modified these theorem statements to emphasize on “any $f$”. The proofs of Theorems 4, 5 and 7 are proved exactly in this way: their basic logic is that, when we keep the first $k$ moments unchanged, any Taylor-expandable function is well approximated, with error starting from the (k+1)th order. This reflects the key universality of our theory: because the compression map is **function-independent**, it simultaneously work for all functions.
>
> Maintaining the value of only one symmetric function is actually trivial; only maintaining the value of many functions makes it meaningful. To see this, let’s say we aim to compress $d$ objects maintaining the value of $f$, which admits a deep-set representation $f(w_1, \dots, w_d) = h(\sum_{i=1}^d g(w_i))$, where $d_{lat} = \operatorname{dim} \operatorname{Image}(g)$ is called the latent dimension. Then we can compress the variables into only $d_{lat}+1$ weighted objects, by Caratheodory’s theorem (the reasoning is similar to Algorithm 2; put it simple, if $d_{lat}=1$, apparently we can choose two $w$'s so that their weighted average is equal to the average of all $g(w_i)$’s). Therefore, we can compress any number of objects into O(1) objects, inducing no error at all.
>
> We believe clarifying that Theorems 4, 5 and 7 work for virtually any symmetric function automatically resolves your concern on that compression doesn’t work for the test loss. Let us explain why compression theory works for test loss in detail. For both compressing dataset and compressing NN parameters, the following argument applies.
>
> (1) Generally, the prediction $\hat{y}(x) = f_x(X, \theta)$ is determined by the training dataset $X$ and the NN parameters $\theta$. As explained in Section 2, $\hat{y}$ is a symmetric function in terms of $X$, and also independently in terms of $\theta$. Here, we use a collective symbol $\hat{y}$ to stand for all possible predictions the model can make, which can have many dimensions.
>
> (2) The test loss function generally looks like $L = \sum_{x \in Test\ Set} \ell(y(x), \hat{y}(x))$, where $y$ stands for ground truth, $\hat{y}$ stands for the prediction. No matter what the test set is, and no matter $\ell$ and $y_a$ specifically is, $L$ is a symmetric function in terms of both $X$ and $\theta$, because $L$ depends on $X$ and $\theta$ solely through $\hat{y}$ (which is a symmetric function).
>
> (3) Since the test loss is a symmetric function of the training dataset, one can train on the compressed training dataset to obtain almost the same test loss. Similarly, since the test loss is a symmetric function of the model parameters (either before or after training is OK), the original model and the compressed model must have the test loss (within our proved bound).

---

### Author Response · Authors · 2025-11-26
**Statement on the significance of our work; list of changes in response to reviewers**

We thank all reviewers for the constructive feedback. To each reviewer, we reply to every weakness and question in detail below. Here, we summarize our main contribution, which seems to be misunderstood, and the changes we have made to improve the manuscript. We also point out criticisms that we feel are unfair. After revision, we believe we have addressed all the concerns of the reviewers.

We would like to emphasize that our contribution is theoretical, and establishes this key, universal result: there exists a compression map from many objects to a polylog-size subset of objects, which keeps the value of almost all symmetric outputs unchanged. Such a universal result is important, unprecedented and difficult, so is worthy of acceptance on its own. It could also have major applications across many subfields of AI. We also do our best to demonstrate our theory in experiment and all our experiments are designed to illustrate our theory, NOT for demonstrating its practical efficiency advancements. While we do believe future works will establish practical relevance of our work, our current manuscript makes zero claims about practice and it is beyond the scope of our work to propose or establish a practical algorithm. In fact, in our manuscript we stated clearly that “The central contribution of our theory is a proof of concept that it is theoretically possible to strongly compress neural networks and datasets, enabling far more efficient use of data and parameters. An important future direction is therefore to develop practical compression algorithms that can improve neural scaling laws at scale.” However, the most common criticisms we receive are about the practicality of the algorithm we used to evaluate the theory, which we feel is unfair. In fact, it is not quite reasonable to evaluate our moment-matching compression method, which we merely intend to show the existence of universal compression, with gauges like performance or computational efficiency improvements.

We invite the reviewers to ask additional questions and help us further improve the manuscript.

**List of changes:**

1. Replaced the proof of the optimality of $\log^m (d)$ in Appendix B. The new proof is self-consistent and does not rely on any conjecture. (related to questions from Reviewers YxjE, ui4S and LiWj)
2. The new Appendix F on the permutation symmetry in transformers, and how to apply compression to transformers. (related to questions from 1dtD and YxjE)
3. The new Appendix G on numerically showing compression to polylog(d). (related to questions from YxjE and ui4S)
4. Replaced $\omega(d)$ by $\varepsilon(d)$ to avoid ambiguity. Added examples ($\varepsilon(d) = d^{-\alpha}$ ) to help understand the results of Theorems 4 and 7 (related to questions from LiWj)
5. Fixed the broken reference in line 190: “Appendix ??” -> “Appendix D” (related to questions from YxjE, 1dtD and ui4S)
6. Other minor revisions.

**New numerical experiments:**

1. Compressing an attention module and showing LTH-like consistency (Appendix F) (related to questions from Reviewers 1dtD and YxjE)
2. Compression algorithm runtime vs d (Appendix D) (related to questions from Reviewer 1dtD)
3. Demonstration of the ability to compress d to log(d) when $m=1$ or $2$ (Appendix G) (related to questions from Reviewers YxjE and ui4S)

---

### Author Response · Authors · 2025-12-03
**Final remark to Area Chair**

Dear Area Chair,

Due to the sudden change of the review process on 11/28/2025, we provide the following remark in respect to the reviewer’s effort in assessing and helping us improve this paper.

### Summary of reviewers’ assessments

- **YxjE:** The original rating was 4. As of 11/27, we were still discussing and [awaiting the reviewer's pending response](https://openreview.net/forum?id=vxkzW4ljeX&noteId=AC7kUpBA5x). We were [clarifying major points](https://openreview.net/forum?id=vxkzW4ljeX&noteId=uqlrlNQhWM) (especially on the universality of our compression method) which the reviewer has misunderstood. We are confident that we have provided conclusive answer to each of the reviewer’s concerns, which either clarified the reviewer’s misunderstanding or improved the soundness and practicality with new numerical experiments (compressing transformers & lossless compression to polylog).
- **1dtD:** The original rating was 6, and the reviewer keeps their rating after one round of discussion. The reviewer mainly asked for extending our compression theory to transformers and a runtime benchmark. We added a new Appendix to theoretically describe compression of transformers, and numerically demonstrated that the dynamical lottery ticket hypothesis holds for multi-head attentions. We added runtime benchmark in Appendix D as well.
- **ui4S:** The original rating was 8, and the reviewer was satisfied by our response and decided to keep the rating. Especially, we added a new numerical demonstration of compressing $d$ objects to $\log^m d$ in response to the reviewer’s question Q1, along with clarification on all other questions.
- **LiWj:** The original rating was 4; as of 11/27, the reviewer has [**raised the rating to 6**](https://openreview.net/forum?id=vxkzW4ljeX&noteId=GH7pp8X8Q2). The main reason that led to this increase is likely that we successfully clarified all points which the reviewer seemed to had overlooked or misunderstood.

### Our contributions

Please see [Statement of the significance of our work; list of changes in response to reviewers](https://openreview.net/forum?id=vxkzW4ljeX&noteId=18r56RmNrC). Our paper accomplishes an important, unprecedented and difficult progress on quantitative universal compression theory in machine learning. As many reviewers noted, our paper stands out by its mathematical rigor and wide applicability. We proposed a concrete compressing algorithm, originally intending only to provide a proof of principle that an optimal compression mapping exists. Also as many reviewers noted, our results could be more practical if it scales better with data dimension and if we show more numerical demonstration. We delivered these numerical demonstrations in the revised paper, and proved that its scaling is optimal. In the revised paper, we strengthened our work by

(1) explaining and demonstrating that compression applies to transformers

(2) completing a self-contained proof that $d\to \log^m d$ compression is optimal

(3) demonstrating that the optimal compression $d\to \log^m d$ is achievable

(4) providing runtime benchmark.

Finally, we would like to express gratitude to all who helped us improve the paper in the rebuttal phase, and especially to the Area Chair for reviewing our paper in this challenging time. We hope you can give fair final assessment in respect to the reviewer’s opinions and our effort in improving the soundness and practicality of our universal compression theory.

---

### Meta-Review · Area_Chair_y2EK · 2025-12-10

**Summary:**

This paper develops a theoretical framework for compressing neural networks and datasets by exploiting permutation symmetry. It proves a universal compression theorem showing that symmetric functions can be roughly approximated with polylogarithmic size (though exponential dependency on the embedding dimension). The same framework implies stronger scaling laws, turning power-law relationships between performance and size into stretched-exponential ones. The authors support their theory with a moment-matching algorithm and small-scale numerical experiments.

Reviewers found the work mathematically sound and clearly written, but raised significant concerns about practicality. The initial experiments are limited to low-dimensional settings, so that the implications to higher-dimension settings were unclear. Indeed, several reviewers questioned the impact of dimensionality on the theoretical bounds, the lack of additional comparisons to standard compression methods, and the limited discussion of how assumptions like smoothness or symmetry apply to modern architectures. The revised version of the document conducted additional experiments, partially addressing concerns about practicality. Additionally, a lower bound was included showing that the polylogarithmic compression rate with exponential dependency on the dimension is optimal.

Therefore, I believe the main remaining concern is that the current theoretical results do not quite apply to deep neural networks, which are complex compositions of many layers, often with non-linear activation functions (ReLU, GeLU, etc.) and non-symmetric components, and thus it is not entirely accurate to compare the theoretical results to end-to-end lottery ticket hypotheses. However, I believe this is nevertheless a good start toward this direction and recommend acceptance.

Note: there are some editorial internal comments, e.g., at the bottom of Page 1, that must be resolved before the final version.

**Reviewer Concerns:**

I believe reviewer concerns about experimental evaluations were addressed by additional experiments and reviewer concerns about the exponential dependency on the dimension are addressed by the lower bound. Concerns about assumptions for the underlying function, e.g., smoothness, may still be outstanding.

**Reviewer Scores:**

Reviewer LiWj explicitly stated that their score would be updated. On the other hand, Reviewer YxjE mentioned additional concerns after one round of interaction during the discussion phase. Reviewer 1dtD and Reviewer ui4S stated they would maintain their score.

---

### Decision · Program_Chairs · 2026-01-26

Accept (Poster)